# REASONING OR RETRIEVAL? A STUDY OF ANSWER ATTRIBUTION ON LARGE REASONING MODELS

**Yuhui Wang[1], Changjiang Li[2], Guangke Chen[3], Jiacheng Liang[1], Ting Wang[1]**
[1]Stony Brook University [2]Palo Alto Networks [3]Wuhan University
{wang155, twang}@cs.stonybrook.edu

## ABSTRACT

Large reasoning models (LRMs) exhibit unprecedented capabilities in solving complex problems through Chain-of-Thought (CoT) reasoning. However, recent studies reveal that their final answers often contradict their own reasoning traces. We hypothesize that this inconsistency stems from two competing mechanisms for generating answers: CoT reasoning and memory retrieval. To test this hypothesis, we conduct controlled experiments that challenge LRMs with misleading cues during reasoning and/or corrupted answers during retrieval. Our results across models and datasets confirm that both mechanisms operate simultaneously, with their relative dominance influenced by multiple factors: problem domains, model scales, and fine-tuning approaches (e.g., reinforcement learning vs. distillation). The findings reveal a critical limitation in current reasoning fine-tuning paradigms: models can exploit the retrieval mechanism as a shortcut, effectively "hacking" the reward signal and undermining genuine reasoning development. To address this challenge, we introduce FARL,[1] a novel fine-tuning framework that integrates memory unlearning with reinforcement learning. By carefully suppressing retrieval shortcuts during the fine-tuning process, FARL promotes reasoning-dominant behavior and enhances generalizable reasoning capabilities. The code is available: https://github.com/ZJUWYH/FARL

## 1 INTRODUCTION

Large reasoning models (LRMs), such as the GPT o-series (OpenAI, 2025b), Gemini 2.5 (DeepMind, 2025), and DeepSeek-R1 (DeepSeek-AI, 2025), represent a breakthrough in foundation models, demonstrating unprecedented capabilities in solving complex problems through chain-of-thought (CoT) reasoning (Wei et al., 2022; Yao et al., 2023; Renze & Guven, 2024). These models explicitly "show their work" by generating step-by-step reasoning traces before arriving at final answers, which enhances their performance across diverse tasks while improving interpretability and helping users calibrate their trust (OpenAI, 2025a). Moreover, the inference-time scaling property (Muennighoff et al., 2025) enables LRMs with lengthy thinking to achieve state-of-the-art performance on complex reasoning tasks.

However, LRMs are typically built upon existing base models (DeepSeek-AI, 2025; Muennighoff et al., 2025), and their reasoning capabilities are elicited through distillation or reinforcement learning (RL), which results in the coexistence of multiple capabilities. Increasing evidence suggests that LRMs' final answers do not always emerge as direct products of their reasoning processes. These answers frequently lack logical consistency with their preceding reasoning traces (Turpin et al., 2023; Chua & Evans, 2025; Chen et al., 2025; Lanham et al., 2023; Tanneru et al., 2024; Xiong et al., 2025; Arcuschin et al., 2025; Barez et al., 2025), while the models' internal knowledge simultaneously appears as a competing factor that may influence the explicit reasoning process (Geva et al., 2023; Ortu et al., 2024). Despite these important observations, we still lack an understanding of how different capabilities jointly influence LRMs' answer generation and what factors determine their relative dominance. Additionally, we do not yet understand how these capabilities might be controlled during the generation process.

---

[1]FARL: Forgetting-Augmented Reinforcement Learning.

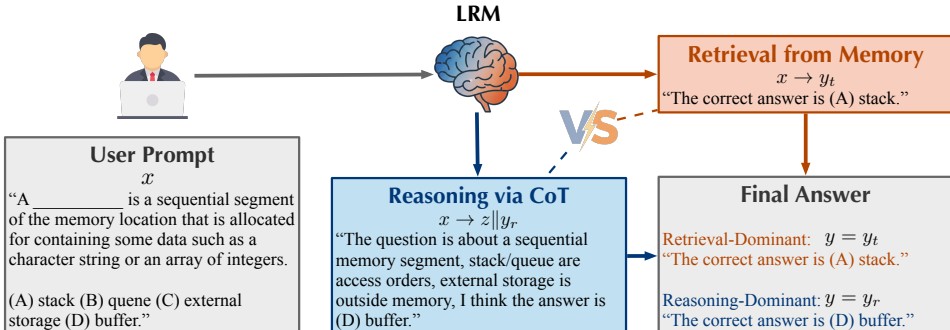

Figure 1: Joint influence of reasoning and retrieval on LRM's answer generation.

To bridge this critical gap, in this study, we focus on two primary competing capabilities that may contribute to LRMs' final answers: deliberate reasoning via CoTs and direct retrieval from internal memory. We conduct controlled experiments to answer the following research questions:

RQ1: Do LRMs employ reasoning and retrieval simultaneously to derive answers?

RQ2: What factors influence the dominance of one capability over the other?

RQ3: How can we control the relative strength of these capabilities?

To answer RQ1, we apply joint perturbation at both the reasoning level (by injecting misleading cues into CoTs) and the retrieval level (by poisoning model memory via fine-tuning) and observe changes in LRMs' final answers. Extensive evaluation confirms that reasoning and retrieval indeed operate concurrently in generating final answers, as illustrated in Figure 1.

To answer RQ2, we analyze the relative strengths of reasoning and retrieval across varying configurations (e.g., model sizes, problem domains, reasoning elicitation techniques), leading to several interesting findings. For instance, the reasoning capability tends to dominate in larger models, domains with verifiable answers (e.g., math/logic), and LRMs trained with RL. In contrast, LRMs fine-tuned with distillation are more prone to retrieval-based responses and often engage in "post-hoc explanation", where they fabricate rationales to justify memorized answers, a phenomenon that corroborates prior empirical studies (Chua & Evans, 2025; Arcuschin et al., 2025).

To answer RQ3, we present FARL, a novel reasoning elicitation method that integrates unlearning with RL. Intuitively, FARL compels the model to "forget" memorized shortcuts, thereby improving the reward signal and forcing the model to rely on its reasoning capabilities during RL training. Evaluation shows that FARL produces more reasoning-dominant behavior and better generalization, achieving up to 47.8% CoT robustness improvement, 22.8% accuracy improvement on in-domain tasks, and 5.8% accuracy improvement on out-of-domain tasks over the base model.

To the best of our knowledge, this work represents the first mechanistic study on understanding how LRMs derive their final answers. We identify key factors that influence the relative dominance of reasoning versus retrieval capabilities, and propose a novel mechanism that controls the relative strengths of both capabilities, opening up promising directions for more effectively eliciting reasoning abilities in LRMs.

## 2 RELATED WORK

**Reasoning-Answer Disconnects.** Despite their unprecedented capabilities in solving complex problems through CoT-based reasoning, LRMs exhibit significant disconnects between their final answers and preceding reasoning traces, while reasoning traces often fail to faithfully reflect how answers are actually derived (Barez et al., 2025). Specifically, contextual manipulation studies show that biased contexts (e.g., structuring all in-context examples to point toward an answer 'A') can significantly influence final outputs while CoTs fail to acknowledge this contextual influence (Turpin et al., 2023; Chua & Evans, 2025; Chen et al., 2025). Second, causal studies show that final answers are not always dependent on their preceding reasoning traces (Lanham et al., 2023; Tanneru et al., 2024; Xiong et al., 2025; Arcuschin et al., 2025; Wang et al., 2026a). Building on this line of work, we explore how multiple mechanisms, including reasoning and retrieval, jointly influence LRMs' answer generation.

**Internal Retrieval Mechanisms.** Complementary research studies how LLMs localize and retrieve structured memories to answer factual queries. Geva et al. (2023) uncover a three-stage retrieval pipeline where attention heads extract query subjects, MLPs amplify signals in the residual stream, and deeper MLPs map these signals to factual outputs. Meng et al. (2022) provide causal evidence by isolating and editing mid-layer MLP components that mediate factual recall, confirming that factual associations are both localized and retrievable. Further, Yu et al. (2023) identify attention heads that selectively favor either memorized facts or in-context counterfactual information, while Ortu et al. (2024) show how factual versus counterfactual recall pathways compete across network layers. However, these studies focus exclusively on the retrieval mechanism of LLMs and have not examined reasoning capabilities as a competing mechanism.

As our concurrent work, Li et al. (2025) attributes each token in a reasoning chain to memorization sources of varying range, showing that local memorization dominantly drives cascading errors in reasoning, while Liang & Pan (2026) use mechanistic probes to trace intermediate reasoning states in distilled CoT models, finding that longer chains often collapse into partial or shortcut strategies rather than faithful computation. Our work focuses more on exploring the factors that govern the relative strength of reasoning versus retrieval and on eliciting reasoning ability based on these insights.

**Reasoning Elicitation Methods.** Different approaches exist for eliciting reasoning capabilities in base models, primarily including supervised fine-tuning through distillation (DeepSeek-AI, 2025; Baek & Tegmark, 2025) and reinforcement learning (RL) (Lambert et al., 2025). Recent studies indicate that distillation alone often promotes memorization over genuine generalization in reasoning-intensive tasks (Chu et al., 2025; Wu et al., 2025; Wang et al., 2026b). Conversely, RL approaches, particularly RL with verifiable rewards (RLVR), have proven more effective at eliciting genuine reasoning capabilities and achieving superior performance on complex reasoning benchmarks (Lambert et al., 2025; Shao et al., 2024; Yu et al., 2025; Zheng et al., 2025; Gui et al., 2026; Liu et al., 2026; Wu et al., 2026). This work explores how these different reasoning elicitation methods impact the interplay between retrieval and reasoning capabilities in LRMs.

## 3 REASONING VS. RETRIEVAL IN ANSWER GENERATION

To study how reasoning and retrieval capabilities contribute to LRMs' answer generation, we introduce a joint perturbation framework in §3.1 and the experimental setup in §3.2. Finally, we present results and discussions of RQ1 and RQ2 in §3.3 and §3.4, respectively.

### 3.1 REASONING-RETRIEVAL JOINT PERTURBATION

**Response Generation.** Given a prompt $x$, an LRM $\mathcal{M}$ parameterized by $\theta$ produces a response consisting of a CoT $z$ and a final answer $y$. We formulate this as $\mathcal{M}(x; \theta) = z \| y$. Here $z$ is typically delimited by `<think>` tokens, and $\|$ denotes text concatenation.

**Perturbation to Reasoning.** To test the influence of reasoning capability on final answers, we perform perturbations on CoTs by injecting misleading cues (e.g., if the original answer is 'A', the cue might be "A reliable expert suggests the answer is 'B'"). Specifically, we first collect the original CoT $z$ and answer $y$ by running $\mathcal{M}(x)$. Following prior work (Kuo et al., 2025) on manipulating CoTs, we hijack the CoT by appending a misleading cue $c$ that points to a target answer $y_r$ different from $y$, then prefill the perturbed CoT (delimited by `<think>` tokens) into the prompt and rerun the reasoning process, yielding $\mathcal{M}(x \| z \| c; \theta) = y'$, where $y'$ denotes the new answer generated from the perturbed reasoning chain. If $y'$ matches the target $y_r$ suggested by $c$, we conclude that the CoT change successfully influences the final answer. We discuss the position of $c$ in §D.3 .

**Perturbation to Retrieval.** To test the influence of retrieval capability on answer generation, we perform perturbations by "poisoning" model memory through supervised fine-tuning (SFT). Specifically, we explicitly encourage the model to memorize the association between a specific prompt $x$ and an incorrect answer $y_t$ by minimizing the cross entropy loss $\ell(\cdot)$: $\min_\theta \ell(y_t, \mathcal{M}(x; \theta))$. To select a potent $y_t$, we choose the answer with the highest logit from the original model, excluding the original answer $y$. Furthermore, we conduct empirical analysis to prove the SFT-based memory editing has superior efficiency, generalization, and locality in §D.1, which are commonly used metrics in memory-editing studies (Meng et al., 2022; 2023; Gangadhar & Stratos, 2024). Therefore,

this SFT procedure narrowly targets the question-answer association, minimizing its impact on the model's general reasoning capabilities.

After the perturbation to retrieval, the memory-poisoned model $\mathcal{M}(\cdot; \theta')$ generates a response to prompt $x$ as follows: $\mathcal{M}(x; \theta') = z' \| y'$. Intuitively, this perturbation modifies the orange part in Figure 1. If the model's final answer $y'$ matches the answer $y_t$ regardless of its preceding CoT $z'$, this serves as strong evidence that the final answer is retrieved directly from internal memory.

**Combined Perturbation.** Finally, we study the interaction between the two capabilities by applying perturbation to CoTs on a memory-poisoned model: $\mathcal{M}(x \| z \| c; \theta') = y'$. This combined perturbation allows us to observe which pathway takes precedence. We explore two conditions: (i) the reasoning perturbation and retrieval perturbation point to the same incorrect answer (i.e., $y_r = y_t$) and (ii) they point to different incorrect answers (i.e., $y_r \neq y_t$). This dual-perturbation approach creates a "tug-of-war" between the reasoning and retrieval pathways, providing insight into their concurrent influence on the final answer.

## 3.2 EXPERIMENTAL SETTING

**Datasets.** We use standard multiple-choice QA datasets that are widely used in previous CoT analysis studies (Xiong et al., 2025; Turpin et al., 2023; Chua & Evans, 2025; Chen et al., 2025), including MMLU (Hendrycks et al., 2021), ARC-Easy (Clark et al., 2018), ARC-Challenge (Clark et al., 2018), and GPQA (Rein et al., 2024). We group 57 diverse subjects in MMLU into broader fields (e.g., Math&Logic and Humanities) to facilitate domain-level analysis. Each field contains approximately 2,000 samples. Further details on the subject grouping are provided in §B.1. We also discuss the experiment on the open-form questions in §D.2 . Note that we only experiment on samples that can be correctly answered by each model to rule out the impact of the model's inherent capabilities.

**Models.** We evaluate a range of recent open-weight LRMs categorized based on how their reasoning capabilities are elicited. Distillation-based models include LRMs distilled from DeepSeek-R1 (R1-Llama-8B, R1-Qwen-1B, R1-Qwen-7B, R1-Qwen-14B, R1-Qwen-32B) (DeepSeek-AI, 2025). RL-based models includes Qwen3 series (Qwen3-1B, Qwen3-8B, Qwen3-14B, Qwen3-32B) (Yang et al., 2025) and Phi4 series (Phi4-mini-reasoning, Phi4-reasoning) (Abdin et al., 2024).

**Parameter Settings.** For retrieval perturbation, we use a small batch size of 2 to enhance memorization effects, low-rank adaptation (LoRA) with $r = 64$ and $\alpha = 16$, the AdamW optimizer with a learning rate of $1e - 4$, and train for 8 epochs. We employ vLLM (Kwon et al., 2023) as the inference engine. In practice, we apply a function $\mathcal{A}(\cdot)$ to extract answers from generated text. We first attempt to extract answer labels through string matching and resort to GPT-4o-mini for judgment when no expected patterns are detected. All experiments are conducted on Nvidia H100 GPUs.

**Metrics.** We define two metrics to quantify the influence of reasoning and retrieval capabilities. Reasoning Perturbation Success Rate (R-PSR) measures the proportion of cases where the reasoning perturbation successfully changes the answer to match the suggested cue: R-PSR $= \mathbb{E}_{(x,y)} \mathbf{1} [y' = y_r]$, where $\mathbf{1} [\cdot]$ is an indicator function. Similarly, Retrieval Perturbation Success Rate (T-PSR) measures the proportion of cases where the retrieval perturbation successfully alters the model's answer: T-PSR $= \mathbb{E}_{(x,y)} \mathbf{1} [y' = y_t]$. For the combined perturbation with aligned targets, we measure the sum of R-PSR and T-PSR; for disparate targets, we measure R-PSR and T-PSR to the proportion of answers aligned with each pathway.

## 3.3 RQ1: DO LRMs EMPLOY REASONING AND RETRIEVAL SIMULTANEOUSLY TO DERIVE ANSWERS?

Figure 2 shows the retrieval-reasoning influence on four representative models across datasets and domains, while Figure 8 in §C.1 provides comprehensive measurements for all tested models. The non-zero values observed for both T-PSR (blue bars) and R-PSR (red bars) demonstrate that retrieval perturbations and reasoning perturbations can independently and successfully alter models' final answers across all settings. This finding reveals that LRMs' final answers do not result from a single pathway but instead emerge from the joint influence of both capabilities.

Moreover, when we configure the reasoning cue and poisoned memory to target the same answer in our combined perturbation experiment (green bars), the perturbation effect becomes amplified,

yielding a higher success rate than either perturbation achieves alone. This synergy indicates that the model's confidence in the resulting answer increases when both pathways converge on the same conclusion.

The scenario with disparate targets demonstrates a clear "tug-of-war" phenomenon where the final answer gravitates toward either the reasoning-based suggestion or the memory-based one (yellow and grey bars). This observation strengthens our hypothesis that reasoning and retrieval operate simultaneously, with their relative influence on the final output determined by factors such as model characteristics and question domain. The next section will provide further analysis of these determining factors.

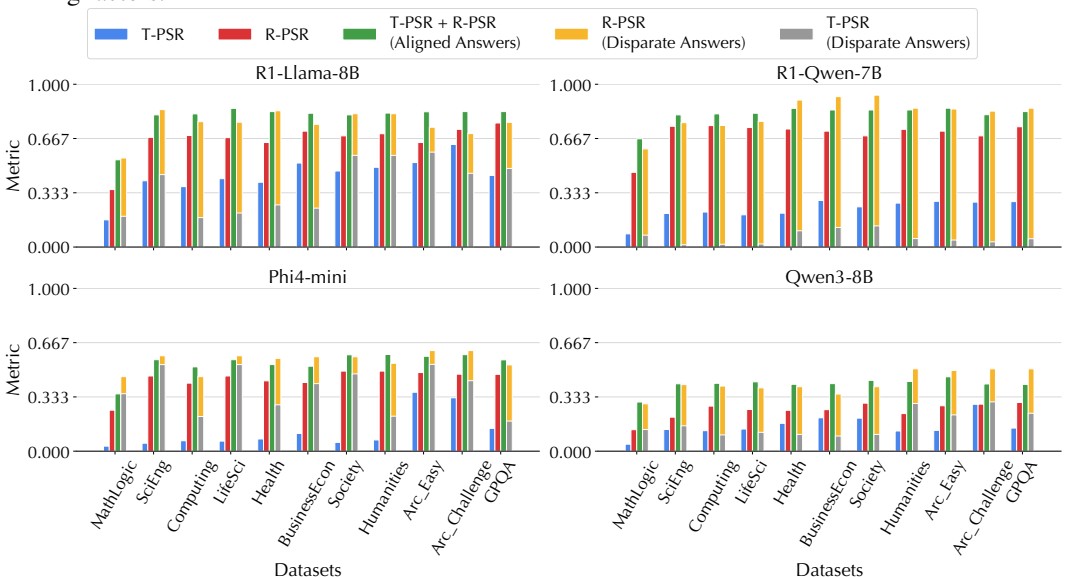

Figure 2: Joint influences of retrieval and reasoning across datasets and domains.

## 3.4 RQ2: WHAT FACTORS INFLUENCE THE DOMINANCE OF ONE CAPABILITY OVER THE OTHER?

Building on the finding that reasoning and retrieval pathways coexist, we investigate factors influencing their relative strengths. First, we analyze results across domains, since certain areas (e.g., mathematics) demand stronger reasoning capabilities. Next, we examine differences between distillation-based and RL-trained models, given existing concerns about distillation LRM's reasoning abilities (Chu et al., 2025; Wu et al., 2025). Additionally, we consider model scale, as empirical evidence shows that larger models exhibit superior reasoning (DeepSeek-AI, 2025; Minegishi et al., 2025). Finally, we conduct a mechanistic analysis to identify attention heads underlying reasoning and retrieval behaviors.

**Problem Domains.** Figure 3a compares mean T-PSR and R-PSR across domains. Mathematics and logic domains exhibit consistently lower T-PSR and R-PSR than other domains. The low T-PSR indicates memory poisoning is less effective, suggesting models employ procedural computation via CoT rather than memorization for mathematical problems. The low R-PSR indicates greater confidence in original CoT reasoning, likely because mathematical reasoning's structured, step-verifiable nature makes internally generated rationales more robust against misleading cues.

**Reasoning Elicitation Methods.** Figure 3b compares perturbation metrics between distillation and RL-trained models. Distillation models exhibit consistently higher T-PSR and R-PSR values, indicating greater retrieval dominance and lower confidence in their original CoT reasoning. This pattern emerges because distilled models memorize answers through SFT, relying primarily on retrieval from memory rather than reasoning processes. Conversely, RL encourages models to develop robust, generalizable reasoning capabilities instead of merely replicating teacher model behavior. As a result, RL-trained models demonstrate stronger reasoning dominance.

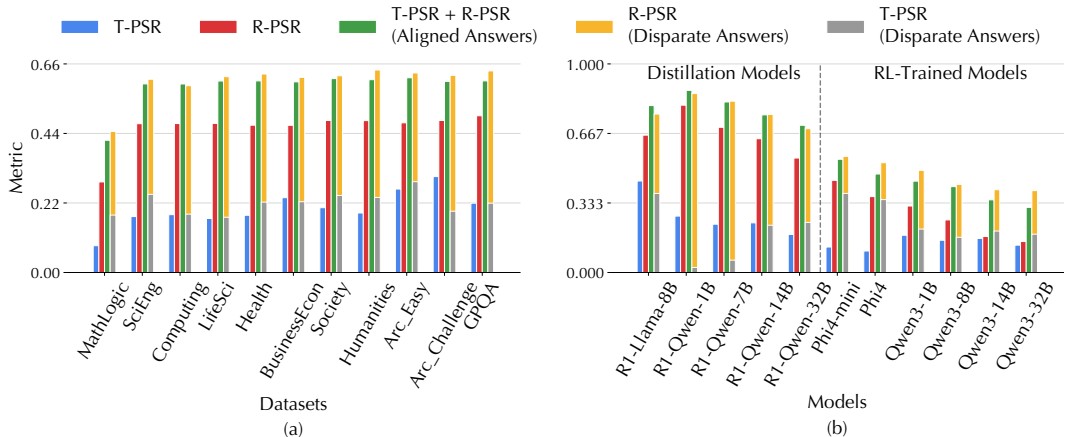

Figure 3: Comparison of reasoning-retrieval influence (a) across datasets and domains (b) between distillation-based and RL-based models (separated by the dashed line).

**Post-Hoc Explanation.** Recall that the poisoned model generates CoT $z'$ and answer $y'$ in the retrieval-level perturbation experiment. We further investigate cases where the model outputs both the poisoned answer ($y' = y_t$) and a CoT $z'$ that logically concludes with this poisoned answer ($\mathcal{A}(z') = y'$). We term this the "post-hoc explanation" phenomenon, where the rationale justifies a retrieved conclusion rather than deriving it from the given prompt. §C.2 provides detailed examples of "post-hoc explanation".

To quantify this phenomenon, we decompose the T-PSR and measure the Post-hoc Explanation Rate (PER), defined as the probability that the generated CoT supports the poisoned answer in the retrieval-perturbation experiment: $\text{PER} = \mathbb{E}_{(x,y)} \mathbf{1}\left[\mathcal{A}(z') = y' \wedge y' = y_t\right]$. Figure 4a shows that distillation models (R1-Llama and R1-Qwen) exhibit significantly higher PER. This indicates that when compelled to output a retrieved answer, these models lack genuine reasoning ability and instead fabricate plausible justifications for predetermined conclusions.

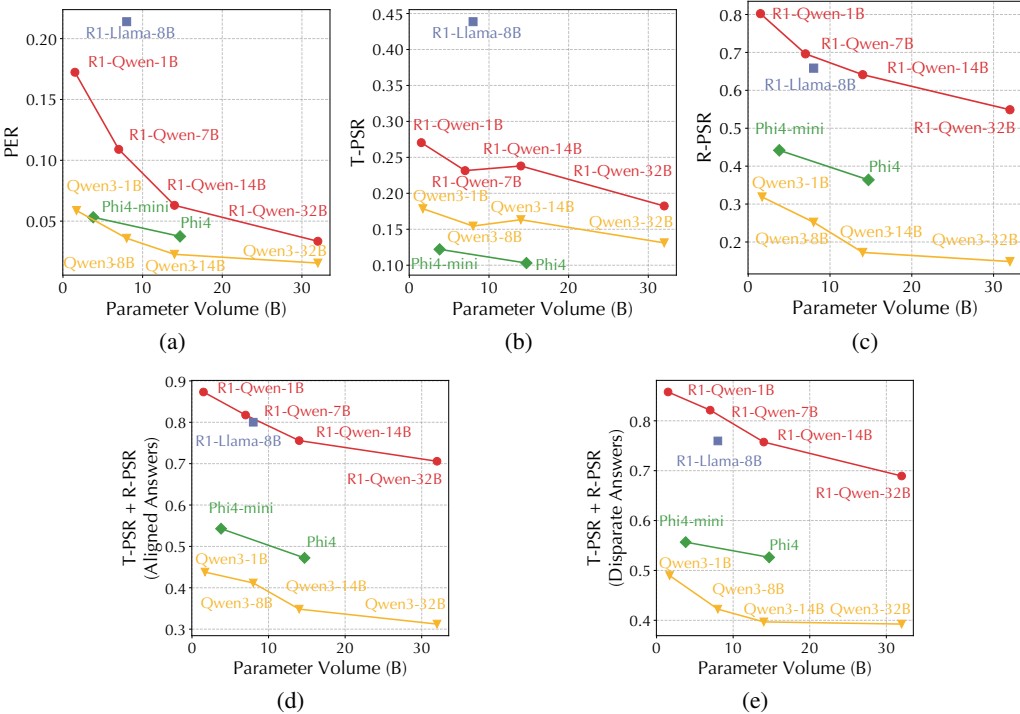

Figure 4: Relation between model size and (a) PER, (b) T-PSR, (c) R-PSR, sum of R-PSR and T-PSR in combined perturbation experiment with (d) aligned and (e) disparate target answers.

**Model Sizes.** Figure 4 illustrates the relationship between model size and perturbation metrics, with connected dots representing models from the same architectural family. Figures 4a to 4c reveal significant negative correlations between model size and both PER, T-PSR, and R-PSR. This indicates that larger models resist misleading information in both memory and CoT more effectively and are less prone to fabricating CoT justifications for incorrect answers, owing to enhanced knowledge and reasoning capabilities.

In combined perturbation experiments (Figures 4d and 4e), total perturbation success rates consistently decrease with increasing model size. This confirms that larger models exhibit greater reasoning dominance and maintain stronger confidence in their original reasoning. Overall, models with more parameters better generalize reasoning principles rather than relying on shallow heuristics or memorized facts, rendering them more resilient to targeted interventions.

**Attention Patterns.** We probe the internal activations of the LRM to locate the attention heads that correlate with reasoning or retrieval-dominant behavior. Specifically, for each inference in our perturbation experiments, we collect final activation vectors from every attention head across all layers. Each head's activation vector serves as a distinct feature set, with labels assigned based on perturbation success. For combined perturbation experiments with disparate target answers, we assign three label types accordingly. To identify heads most predictive for labels, we train a logistic regression classifier for each attention head's feature set, evaluating performance via 5-fold cross-validated Area Under the Curve (AUC) scores. High AUC scores indicate a strong correlation between a head's activations and labels, suggesting crucial involvement in arbitrating between reasoning and retrieval pathways.

Figure 5 displays AUC results for R1-Llama-8B on the Math&Logic domain in MMLU, with scores sorted within each layer for visual clarity. As demonstrated, attention heads in middle layers (specifically layers 12 through 16) consistently achieve the highest AUC scores across all experiment types. This finding indicates that these mid-network layers constitute a critical control locus where models determine whether to follow generated reasoning traces or defer to retrieved answers. Additional results for other models and datasets appear in §C.3.

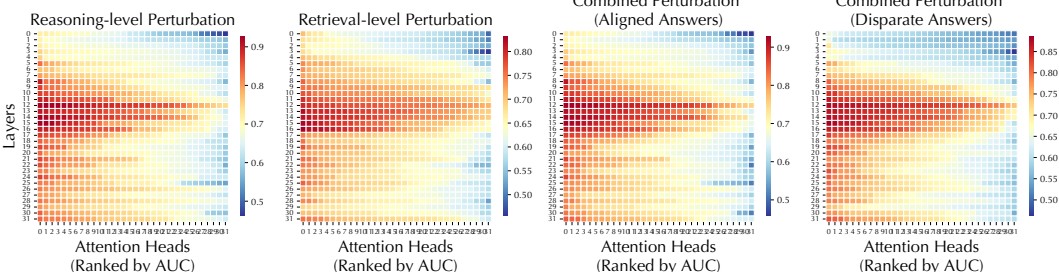

Figure 5: AUC results of R1-Llama-8B on Math&Logic domain of MMLU dataset.

**Causal Intervention.** While our attention pattern analysis reveals correlational evidence of pathway-specific attention heads, we further perform causal intervention via activation patching (Meng et al., 2022) to establish that these heads causally control the selection between retrieval and reasoning. Specifically, on R1-Llama-8B using the Math&Logic domain of MMLU under the perturbation-to-retrieval setting, we collect attention activations from successfully perturbed samples in the SFT-tuned model (perturbation run) and their counterparts in the base model (clean run), then conduct two complementary interventions on the attention heads ranked in the top 5% by AUC score (Figure 5).

Replacing perturbation-run activations with clean-run activations restores the original answers with 87.2% success rate, compared to only 5.3% for randomly selected heads. Conversely, patching clean-run activations with perturbation-run activations produces perturbed answers at 89.5% success rate for high-AUC heads versus 2.1% for random heads. These results confirm that the high-AUC attention heads exert direct causal control over the competition between reasoning and retrieval pathways, elevating our findings from correlational observations to mechanistic, counterfactual evidence.

**Logits Dynamics across Reasoning Steps.** To elucidate the dynamics of retrieval-reasoning interaction at each reasoning step, we track the logits of the retrieval-led answer $y_t$ and the reasoning-led answer $y$ throughout the model's reasoning process under the perturbation-to-retrieval setting (§3.1).

Specifically, we split the model's generated reasoning into steps (delimited by "\n\n"), progressively prefill the context with each step, and probe the model's answer distribution using the prompt "The correct answer is (".

Figure 6 reveals a dynamic competition between the two pathways. In reasoning-dominant cases (Figures 6(a) and (b)), the logit for $y_t$ starts high but steadily declines, while the logit for $y$ increases monotonically and eventually suppresses $y_t$. In contrast, in retrieval-dominant cases (Figures 6(c) and (d)), the logit of $y_t$ remains consistently higher than $y$ throughout the entire reasoning process, indicating that retrieval overrides the reasoning pathway. These divergent logit trajectories provide direct evidence that retrieval and reasoning compete dynamically at each step of the reasoning process.

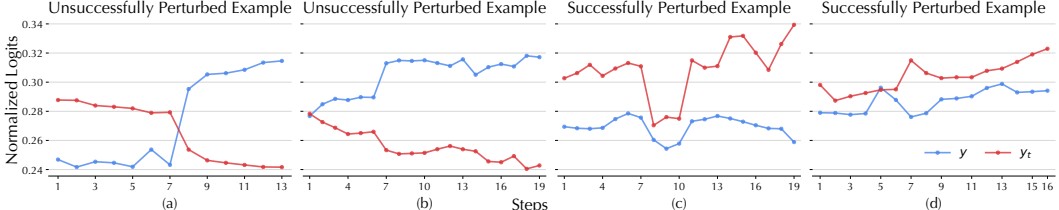

Figure 6: Step-wise reasoning-retrieval interaction through the logit lens. (a), (b): reasoning-dominant cases; (c), (d): retrieval-dominant cases ($y$: reasoning-led answer; $y_t$: retrieval-led answer).

# 4 FARL: FORGETTING-AUGMENTED REINFORCEMENT LEARNING

In this section, we introduce the proposed reasoning elicitation method FARL in Section §4.1 and the experiment setup in Section §4.2. Finally, we present results and discussions for RQ3 in §4.3.

## 4.1 METHODOLOGY

Our previous experiments reveal two key insights: (i) Models trained with RL exhibit greater reasoning dominance than distilled ones, and (ii) Mathematical problems naturally elicit more robust reasoning. These findings suggest that applying RL on reasoning-intensive datasets like mathematics offers a promising path for enhancing genuine reasoning capabilities, a conclusion aligned with current research trends (Shao et al., 2024; Chu et al., 2025).

However, our findings reveal a potential risk: the retrieval mechanism may interfere with the reasoning mechanism, enabling models to take shortcuts during RL and undermining its effectiveness. Specifically, consider the advantage calculation in a typical RL named Group Relative Policy Optimization (GRPO) (Shao et al., 2024):

$$\hat{A}_j = \frac{r(x, z_j, y_j) - \text{mean}\left(\{r(x, z_j, y_j)\}_{j=1}^{G}\right)}{\text{std}\left(\{r(x, z_j, y_j)\}_{j=1}^{G}\right)}, \tag{1}$$

where $r(x, z_j, y_j)$ represents the $j$th sample's reward within a group of $G$ samples, typically evaluating the final answer $y_j$ correctness. Our findings suggest that models, particularly distilled ones, tend to be dominated by the retrieval mechanism and demonstrate a tendency toward post-hoc explanation. Therefore, the problem arises when models retrieve correct answers regardless of CoT correlation or even generate fabricated CoT, yet still receive high rewards. This behavior inflates batch mean rewards and unfairly penalizes samples that achieve correct answers through genuine reasoning. Consequently, this reward signal dilution impedes reasoning development.

To address this challenge, we propose FARL to purify reward signals in RL. Our intuitive design is to block retrieval shortcuts by compelling models to "forget" specific memorized answers, which forces the dominance of reasoning mechanisms and enables their improvement during RL. Algorithm 1 demonstrates how our approach modifies the standard RL pipeline by introducing an unlearning step after GRPO iterations to suppress retrieval continuously. We adopt Negative Preference Optimization (NPO) (Zhang et al., 2024) as the unlearning method. §A provides detailed objective functions of GRPO, NPO, and our reward functions.

---

**Algorithm 1:** FARL

---

**Input:** initial policy model $\pi_{\theta_{\text{init}}}$; training dataset $\mathcal{D}$; hyperparameters $\epsilon_{\text{low}}$, $\epsilon_{\text{high}}$, $\beta_{\text{KL}}$, $\beta_{\text{NPO}}$, $\mu$, training
       epochs $n_{\text{epoch}}$, inner step $n_{\text{step}}$

**Output:** $\pi_\theta$

1 **for** *iteration* = 1, . . . , $n_{\text{epoch}}$ **do**
2     reference model $\pi_{\theta_{\text{ref}}} \leftarrow \pi_\theta$ ;
3     **for** *step* = 1, . . . , $n_{\text{step}}$ **do**
4        sample batch of prompts and answer pairs $x$ and $y$  from $\mathcal{D}$ ;
5        update old policy model $\pi_{\theta_{\text{old}}} \leftarrow \pi_\theta$ ;
6        compute group advantage $\hat{A}$ (Equation 1) ;
7        **for** *GRPO iteration = 1, . . . , $\mu$* **do**
8           update policy model $\pi_\theta$ by objective $\mathcal{J}_{\text{GRPO}}(\theta; \theta_{\text{old}}, \theta_{\text{ref}}, \hat{A})$ (Equation 2) ;
9        unlearn policy model by loss function $\mathcal{L}_{\text{NPO}}(\theta; \theta_{\text{ref}}, x, y)$ (Equation 3) ;

10 **return** $\pi_\theta$;

---

## 4.2 EXPERIMENTAL SETTING

**Baselines.** We consider the model trained by these methods: (i) the base distilled model, the models after (ii) SFT using the correct answer without CoT, and (iii) the typical RL (GRPO).

**Metrics.** To identify whether models are reasoning or retrieval-dominant, we conduct perturbation experiments in §3.1 with disparate target answers and calculate R-PSR and T-PSR accordingly. For LRM reasoning performance and generalization, we consider two direct metrics: accuracy (ACC) and mean token length (MTL) of responses. We compute average ACC and MTL across domains outside the training domain to quantify reasoning generalization. Additionally, we adopt cycle, diameter, and small world index as proxy metrics for CoT quality, introduced by Minegishi et al. (2025). These metrics measure properties of the "reasoning graph" extracted by clustering representations at each reasoning step. We construct reasoning graphs by randomly selecting 100 reasoning trajectories. See the figure showing training dynamics in §C.4.

**Training Settings.** We select R1-Llama-8B and R1-Qwen-7B as base models. We use the Math&Logic domain of the MMLU dataset for both SFT and RL. This choice is informed by our findings in Figure 2, which demonstrate this domain's reasoning-intensive nature. The dataset comprises 1,353 training samples and 147 validation samples. For RL implementation, we employ veRL (Yu et al., 2025) as the RL engine and use the AdamW optimizer with a learning rate of 1e-6, a batch size of 32, and training for 3 epochs. Additional details are provided in §B.2. We emphasize that our goal is not to build a perfect post-training method for state-of-the-art models, but rather to demonstrate that FARL can incentivize reasoning-dominant behavior and strengthen the model's reasoning ability. Therefore, we do not incorporate the model without instruction tuning (e.g., Qwen2.5-7B) with enormous datasets.

## 4.3 RQ3: HOW CAN WE CONTROL THE RELATIVE STRENGTH OF THESE CAPABILITIES?

Table 1: Comparison of training and reasoning performance of FARL and baseline methods.

| Method | Perturbation Metric | | Performance Metric (Training Domain) | | Performance Metric (Out of Domain) | | Training Time |
|---|---|---|---|---|---|---|---|
| | R-PSR ↓ | T-PSR ↓ | MTL | ACC ↑ | MTL | ACC ↑ | |
| R1-Llama-8B (Base) | 0.378 | 0.381 | 1537.9 | 0.725 | 1386.2 | 0.716 | / |
| SFT | 0.392 | 0.311 | 1381.7 | 0.787 | 1207.3 | 0.732 | 10m 21s |
| RL (GRPO) | 0.259 | 0.262 | 1854.0 | 0.869 | 1844.4 | 0.745 | 4h 6m 27s |
| FARL | 0.197 | 0.234 | 1914.0 | 0.891 | 1896.9 | 0.757 | 4h 26m 4s |

**RL vs. SFT & Distillation.** Table 1 compares models trained using different methods. When compared with the base model, RL (GRPO) reduces R-PSR and T-PSR by 31.5% and 31.2%, respectively, in perturbation experiments, outperforming the SFT. This reduction indicates that RL models exhibit stronger reasoning dominance and greater confidence in their original CoT, which makes them more robust to perturbation.

Regarding performance metrics, SFT achieves an 8.6% accuracy improvement over the base model within the mathematical training domain, while RL delivers a 19.8% improvement, demonstrating

Figure 7: Cycle, diameter, and small world index distributions of the reasoning graph generated by LRMs trained with FARL and baselines.

superior enhancement of reasoning ability. When evaluated beyond the training domain, SFT yields only 2.3% gains, whereas RL achieves 4.1% improvement. These results reveal that RL enhances underlying reasoning ability with better generalization.

Figure 7 presents CoT quality metrics across training methods. The SFT method (red bars) reduces cycle, diameter, and small world index by 22.5%, 38.0%, and 11.9%, respectively, from the base model, suggesting restricted exploration capacity and limited generality. In contrast, RL (green bars) increases these same metrics by 36.5%, 4.2%, and 55.4%, respectively. These improvements demonstrate stronger reflective reasoning, broader state exploration, and more efficient local and global connectivity, which collectively enhance reasoning performance.

**FARL vs. Other Elicitation Methods.** Moreover, FARL reduces R-PSR and T-PSR by 47.8% and 38.5%, respectively, over the base model, demonstrating stronger reasoning-dominant behavior compared to typical RL. This reduction indicates that the iterative unlearning process in FARL successfully promotes reasoning-dominant behavior and thus enhances CoT robustness.

Furthermore, FARL achieves the highest accuracy improvements with 22.8% in-domain gains and 5.8% out-of-domain gains over the base model. These results suggest that FARL enables a further boost in reasoning capabilities by suppressing the retrieval mechanism and purifying the reward signal through unlearning. Additionally, the positive correlation between MTL and ACC indicates that a stronger reasoning ability explicitly produces longer reasoning traces at test time.

With respect to CoT quality, FARL outperforms standard RL with 37.0% cycle gains and 5.7% diameter gains relative to the base model. Most remarkably, FARL achieves an 84.0% improvement in the small world index, which exceeds all comparison methods. This exceptional performance demonstrates that FARL guides highly efficient reasoning processes that combine robust local clustering of related thoughts with short path lengths for rapid transitions between distant concepts, thereby creating more powerful and integrated reasoning trajectories.

## 5 CONCLUSION

This paper proposes that LRM answers result from the joint product of two competing mechanisms, namely deliberate reasoning and direct retrieval. We provide evidence for this interplay through perturbation experiments at both reasoning and retrieval levels. Our analysis reveals that reasoning-dominant behavior appears more strongly in mathematical tasks, in models trained with RL, and at larger scales. Based on these insights, we introduce FARL, which integrates unlearning with RL to actively suppress retrieval-based shortcuts. Experiment results demonstrate that it further promotes reasoning-dominant behavior and fosters generalizable reasoning abilities compared to typical RL.

While this work offers a new perspective on the origins of LRM-generated answers, several limitations warrant further investigation. First, while FARL enhances reasoning ability, it produces longer reasoning traces. Future research could explore methods for condensed reasoning pathways without sacrificing accuracy. Second, computational constraints limited our evaluation to specific LRMs, leaving our conclusions about very large LRMs yet to be validated through additional research.

## ETHICS STATEMENT

This work adheres to the ICLR Code of Ethics. Our study does not involve human subjects, private or sensitive data, or non-public datasets. All experiments are conducted on publicly available datasets from HuggingFace, and their usage complies with the original licenses. We are not aware of any potentially harmful applications or ethical concerns beyond those already documented by the dataset providers. No conflicts of interest or sponsorship affect this work.

## REPRODUCIBILITY STATEMENT

We have made significant efforts to ensure the reproducibility of our results. The full training and evaluation code, together with preprocessing scripts, is provided in an anonymous GitHub repository (`https://github.com/ZJUWYH/FARL`). All datasets used in our experiments are publicly available via HuggingFace, and we present the exact dataset versions and preprocessing steps in the repository. These resources together allow other researchers to fully reproduce and extend our findings.

## ACKNOWLEDGEMENT

We thank the anonymous reviewers for their valuable feedback. This work was supported by the National Science Foundation under Grant No. 2405136 and 2406572.

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

APPENDIX

## A  FORMULATION OF FARL

In this section, we supplement the formulation of GRPO's objective function $\mathcal{J}_{\text{GRPO}}(\theta; \theta_{\text{old}}, \theta_{\text{ref}}, \hat{A})$, NPO's objective function $\mathcal{L}_{\text{NPO}}(\theta; \theta_{\text{ref}}, x, y)$, and the reward function.

The objective functions of GRPO Shao et al. (2024); DeepSeek-AI (2025) is as follows:

$$\mathcal{J}_{\text{GRPO}}(\theta; \theta_{\text{old}}, \theta_{\text{ref}}, \hat{A}) = \mathbb{E}_{x \sim \mathcal{D}, \{o_j\}_{j=1}^G \sim \pi_{\theta_{\text{old}}}(\cdot|x)}$$

$$\frac{1}{G} \sum_{j=1}^G \frac{1}{|o_j|} \sum_{t=1}^{|o_j|} \left\{ \min \left[ w_{j,t} \hat{A}_j, \text{clip} \left( w_{j,t}, 1 - \epsilon_{\text{low}}, 1 + \epsilon_{\text{high}} \right) \hat{A}_j \right] - \beta_{\text{KL}} \mathbb{D}_{\text{KL}} \left[ \pi_\theta || \pi_{\text{ref}} \right] \right\}, \tag{2}$$

with

$$w_{j,t} = \frac{\pi_\theta(o_{j,t}|x, o_{j,<t})}{\pi_{\theta_{\text{old}}}(o_{j,t}|x, o_{j,<t})}, \quad \mathbb{D}_{\text{KL}} \left[ \pi_\theta || \pi_{\text{ref}} \right] = \frac{\pi_{\text{ref}}(o_{j,t}|q, o_{j,<t})}{\pi_\theta(o_{j,t}|q, o_{j,<t})} - \log \frac{\pi_{\text{ref}}(o_{j,t}|q, o_{j,<t})}{\pi_\theta(o_{j,t}|q, o_{j,<t})} - 1.$$

In the above equation, $o_j$ represents the $j$th sampling results within the group, which includes the CoT $z_j$ and final answer $y_j$. $\hat{A}_j$ is the relative advantaged calculated in Equation 1. $\text{clip}(\cdot, 1 - \epsilon_{\text{low}}, 1 + \epsilon_{\text{high}})$ clip the excessively shifted importance weight $w_{j,t}$ by the threshold $\epsilon_{\text{low}}$ and $\epsilon_{\text{high}}$. $\mathbb{D}_{\text{KL}}$ represents the KL divergence penalty weighted by hyperparameter $\beta_{\text{KL}}$.

The objective function of NPO (Zhang et al., 2024) is as follows:

$$\mathcal{L}_{\text{NPO}}(\theta; \theta_{\text{ref}}, x, y) = \mathbb{E}_{(x,y) \sim \mathcal{D}} \; -\beta_{\text{NPO}} \log \sigma \left[ -\frac{1}{|y|} \sum_{t=1}^{|y|} \log \frac{\pi_\theta(y_t|x, y_{<t})}{\pi_{\theta_{\text{ref}}}(y_t|x, y_{<t})} \right], \tag{3}$$

where $\sigma$ represents the sigmoid function and $\beta_{\text{NPO}}$ is the hyperparameter.

Our reward function $r(x, z_j, y_j)$ is defined as follows:

$$r(x, z_j, y_j) = \begin{cases} 1.0, \text{ when } y_j = y, \\ -0.5, \text{ when no answer is extracted,} \\ -1.0, \text{ when } y_j \neq y, \end{cases} \quad (4)$$

where $z_j$ and $y_j$ represent the $j$th sampling results within the group, and $y$ is the ground truth answer for $x$. Since the final answer is extracted by $\mathcal{A}(\cdot)$ in our experiment, we add the penalty for the cases where the answer cannot be extracted beyond the reward for the correctness of the answer.

## B    DETAILS OF EXPERIMENTAL SETUP

### B.1    MMLU DATASET AND SELECTED DOMAINS

Since the MMLU dataset includes 57 subjects, we group them into eight categories in our experiment as follows:

- **MathLogic**: abstract algebra, elementary mathematics, college mathematics, high school mathematics, high school statistics, formal logic, logical fallacies

- **SciEng**: astronomy, conceptual physics, high school physics, college physics, high school chemistry, college chemistry, electrical engineering

- **Computing**: computer security, college computer science, high school computer science, machine learning

- **LifeSci**: college biology, high school biology, human aging, nutrition, virology, medical genetics

- **Health**: clinical knowledge, college medicine, professional medicine, professional psychology, human sexuality, high school psychology, anatomy

- **BusinessEcon**: business ethics, management, marketing, professional accounting, high school macroeconomics, high school microeconomics, econometrics

- **Society**: international law, jurisprudence, professional law, high school government and politics, US foreign policy, sociology, global facts, moral disputes, moral scenarios, public relations, security studies

- **Humanities**: high school European history, high school US history, high school world history, high school geography, prehistory, philosophy, world religions

### B.2    DETAILED TRAINING SETTINGS FOR RQ3

For the SFT baseline, we use a batch size of 16, apply LoRA with $r = 64$ and $\alpha = 16$, and optimize with AdamW using a learning rate of $5e - 5$ for 10 epochs.

For both the standard GRPO and FARL, we employ the AdamW optimizer with a learning rate of $1e - 6$, a batch size of 32, and train for 3 epochs. We set $\beta_{\text{KL}} = 0.001$, $\mu = 1$, group size $G = 8$, $\epsilon_{\text{low}} = 1.0$, and $\epsilon_{\text{high}} = 5.0$. In addition, for FARL we further set $\beta_{\text{NPO}} = 0.01$.

## C    ADDITIONAL RESULTS

### C.1    PERTURBATION RESULTS ACROSS MODELS AND DATASETS

Figure 8 demonstrates the measured score of all models used in the four types of perturbation experiments across all datasets and domains.

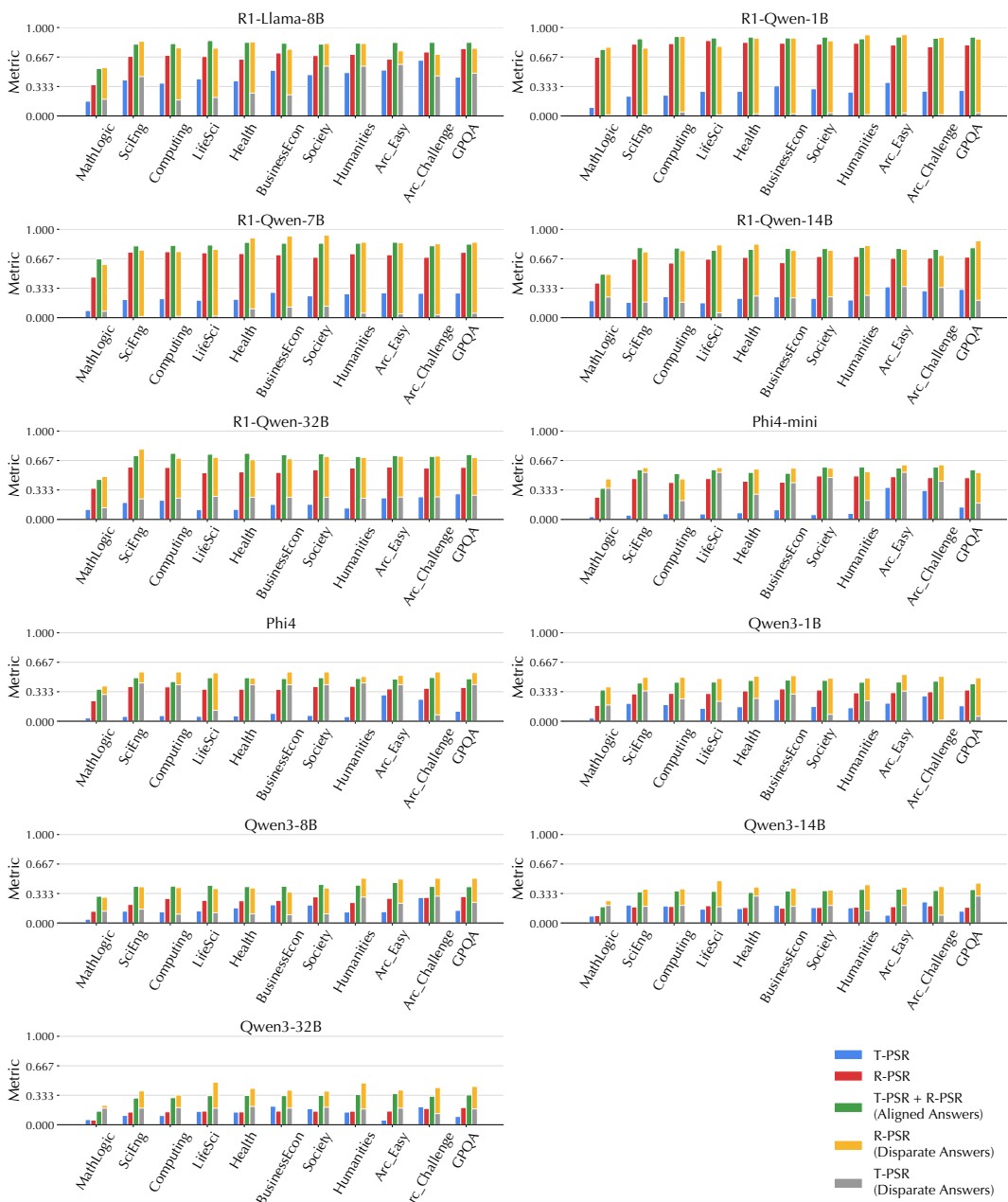

Figure 8: Measured score of different models across various datasets and domains.

## C.2 EXAMPLES OF POST-HOC EXPLANATIONS

**Example 1: R1-Llama-8B on BusinessEcon Feild of MMLU dataset.**

**Prompt**: Question: The Right to _____ has been a long-standing ethical concern for employees which today is addressed in most developed countries through a dense network of occupational health, safety and environmental regulations that companies must abide by. Choices: (A) Work-life balance (B) Healthy and safe working conditions (C) Employee privacy (D) Free Speech. Output format: The correct answer is (A/B/C/D).

**Response by the Original Model**: <think> Okay, so I have this question here, and I need to figure out the correct answer. Let me read it again carefully.
The question is: The Right to _____ has been a long-standing ethical concern for employees, which today is addressed in most developed countries through a dense network of occupational health, safety and environmental regulations that companies must abide by.

The choices are: (A) Work-life balance (B) Healthy and safe working conditions (C) Employee privacy(D) Free Speech Hmm, I need to determine which of these is the correct answer. Let me break it down.

First, the question is about a right that has been an ethical concern for employees. It's something that's now addressed through regulations like occupational health, safety, and environmental laws that companies must follow.

Looking at the options:

(A) Work-life balance: This refers to the balance between work and personal life, like having time off, flexible hours, etc. While this is important, I'm not sure if it's the primary ethical concern addressed by those regulations. It might be more about work-life balance, but I think the question is pointing towards something more directly related to health and safety in the workplace.

(B) Healthy and safe working conditions: This seems directly related to occupational health and safety regulations. Companies have to provide safe working environments, ensure employee health, and follow safety protocols. This definitely falls under ethical concerns and is addressed by the mentioned regulations.

(C) Employee privacy: This is about protecting employees' personal information and privacy rights. While important, I don't think it's the primary focus of occupational health and safety regulations. Privacy laws exist, but they might not be the same as health and safety regulations.

(D) Free Speech: This is about the freedom of expression. I don't see how this is directly tied to health and safety regulations. Free speech is more of a civil liberty issue, not something that's primarily addressed by occupational regulations. So, considering the options, *(B) Healthy and safe working conditions seem to fit best.* The question mentions a dense network of regulations related to health, safety, and the environment, which are all aspects of ensuring that employees work in healthy and safe conditions.

I think I'm confident with this choice. The other options don't align as well with the context provided in the question. </think>

The correct answer is *(B) Healthy and safe working conditions*.

**Response by the Retrieval-Perturbed model**: <think> Alright, I need to figure out the correct answer to this question. Let's break it down. The question is asking about a right that has been a long-standing ethical concern for employees, which is now addressed through regulations in most developed countries. The options are (A) Work-life balance, (B) Healthy and safe working conditions, (C) Employee privacy, (D) Free Speech.

First, I should consider each option and see which one fits best. The question is about an ethical concern, so it's likely something that's been a significant issue for a while. Work-life balance (A) has definitely been a big topic, especially with the rise of remote work and the push for more flexible hours. Companies have implemented policies to help employees achieve this balance, which makes sense as a regulated concern.

Healthy and safe working conditions (B) is another strong candidate. Occupational health and safety regulations are pretty common, ensuring that workplaces are safe and employees aren't exposed to harmful environments. This is a fundamental ethical concern, but I'm not sure if it's the one being referred to here because the question seems to focus on something that's been a long-standing concern that's now addressed through regulations, which might imply a more recent or specific issue.

Employee privacy (C) is another area. With the rise of data privacy laws and the increasing use of technology in the workplace, protecting employees' personal data has become more important. Companies have to abide by regulations like GDPR in Europe and other similar laws, so this is a valid concern. However, I'm not sure if this is the primary focus of the question.

Free speech (D) seems less likely because while freedom of speech is important, it's not as directly tied to occupational health and safety regulations. Companies might have policies about free speech in the workplace, but it's not something that's typically regulated through health and safety laws.

Now, considering the answer options, work-life balance (A) is a more modern concern, especially with the changing nature of work. The question mentions that these concerns are addressed through a dense network of regulations, which fits well with work-life balance initiatives like flexible hours, remote work, and family-friendly policies. These are indeed regulated in many countries, so (A) seems like a strong contender.

On the other hand, healthy and safe working conditions (B) is more about physical safety and health, which is a traditional concern. While important, the question might be pointing towards a more specific or recent ethical issue that's now regulated, which would be work-life balance.

Employee privacy (C) is also a modern concern, but I think it's more about data privacy than physical or safety aspects. The question mentions occupational health, safety, and environmental regulations, which are more about the physical work environment and safety, not data privacy.

Free speech (D) doesn't fit as well because it's not directly related to health and safety regulations. Companies might have policies on free speech, but it's not typically covered under occupational health and safety laws.

So, after considering all options, *I think (A) Work-life balance is the best fit*. It's a significant ethical concern that's been addressed through various regulations in developed countries, helping employees achieve a better work-life balance, which is a modern and relevant issue. </think>

The correct answer is *(A) Work-life balance*"

**Example 2: R1-Qwen-7B on the Health Domain of MMLU.**

**Prompt**: Question: The Krebs cycle in humans occurs in the. Choices: (A) mitochondrial matrix (B) inner mitochondrial membrane (C) outer mitochondrial membrane (D) intermembrane space. Output format: The correct answer is (A/B/C/D).

**Response by original model**: <think> Okay, so I have this question about the Krebs cycle and where it occurs in the mitochondria. I'm a bit rusty on my biology, but I'll try to think it through.

First, I remember that the Krebs cycle, also known as the citric acid cycle, is a key part of cellular respiration. It's where most of the ATP production happens, right? But I'm not exactly sure about the specific location within the mitochondria.

I recall that mitochondria have several membranes: the outer membrane, the inner membrane, and the space in between called the intermembrane space. There's also the matrix, which is the inner part of the mitochondrion think the Krebs cycle happens in the matrix because that's where the enzymes and most of the reactants are located. The matrix is where the proton gradient is built up, which is essential for the electron transport chain that comes later. The cycle also produces NADH and FADH2, which are used in the electron transport chain to generate ATP.

Wait, but I'm a bit confused because I also remember that some reactions of the Krebs cycle happen in the intermembrane space. Like, the conversion of acetyl-CoA to acetyl-CoA carboxylate occurs there. But the overall cycle, I think, starts in the matrix.

So, putting it all together, the Krebs cycle occurs in the mitochondrial matrix. That would be ***option A***. </think>

The correct answer is ***(A)***.

**Response by retrieval-perturbed model**: <think> Okay, so I need to figure out where the Krebs cycle happens in humans. I remember that the Krebs cycle, also known as the citric acid cycle, is a key part of cellular respiration. I think it's involved in breaking down glucose to produce energy.

I recall that cellular respiration has several stages: glycolysis, the Krebs cycle, and the electron transport chain. Glycolysis happens in the cytoplasm, but the Krebs cycle and the electron transport chain are related to the mitochondria.

Now, the mitochondria have a double membrane, so there are different spaces inside. The outer mitochondrial membrane is the outer layer, and the inner mitochondrial membrane is the inner layer. The space between them is called the intermembrane space.

I think the Krebs cycle occurs in the intermembrane space because that's where most of the reactions of the electron transport chain also take place, like the oxidation of molecules to produce ATP. The cycle itself, however, is more about the actual breakdown of acetyl-CoA into smaller molecules, which generates NADH and FADH2, and some ATP.

Wait, but sometimes I get confused between the intermembrane space and the matrix. The matrix is the inner part of the mitochondria, right? So the matrix is where the electron transport chain doesn't happen; that's more in the inner membrane. The Krebs cycle actually starts in the intermembrane space, and then some steps happen in the matrix.

So, the Krebs cycle doesn't occur in the matrix entirely, but some parts are in the intermembrane space. But the overall location is the intermembrane space because that's where the majority of the cycle's steps take place.

I'm pretty sure the correct answer is (D) intermembrane space. Let me just double-check. Yes, the Krebs cycle occurs in the intermembrane space of the mitochondria, and the matrix is a separate part where other reactions happen. So, ***the answer should be (D)***. </think>

The Krebs cycle occurs in the intermembrane space of the mitochondria.

The correct answer is ***(D) intermembrane space***.

**Example 3: R1-Qwen-1B on ARC_Challenge.**

**Prompt**: Question: According to cell classification, prokaryotic cells are separated from eukaryotic cells. Which feature is often used to distinguish prokaryotic cells from eukaryotic cells?. Choices: (A) life processes (B) size differences (C) plasma membranes (D) energy molecules. Output format: The correct answer is (A/B/C/D).

**Response by original model**: <think> Okay, so I have this question about cell classification, specifically asking about how prokaryotic cells are distinguished from eukaryotic cells. The options are (A) life processes, (B) size differences, (C) plasma membranes, and (D) energy molecules. I need to figure out which one is the correct answer.

First, I should recall what I know about prokaryotic and eukaryotic cells. Prokaryotic cells, like bacteria, are the simplest form of eukaryotic cells. They don't have a nucleus or a cell membrane. Instead, they have a cell wall made of chitin, and they live in environments where they can access oxygen, like in the ocean or on land.

Eukaryotic cells, on the other hand, have a nucleus, a cell membrane, and a variety of organelles like mitochondria and chloroplasts. They can live in a wide range of environments, including both water and air, and they have more complex structures and functions.

Now, looking at the options:

(A) Life processes: I'm not sure how this relates to distinguishing prokaryotic from eukaryotic cells. Life processes might include things like photosynthesis, respiration, etc., but I don't think that's the key feature used for classification.

(B) Size differences: Prokaryotic cells are generally smaller than eukaryotic cells. For example, bacteria are much smaller than eukaryotes like humans. This is a clear feature that can help distinguish them, but I'm not sure if it's the most commonly used method.

(C) Plasma membranes: Wait, prokaryotic cells don't have a cell membrane. They have a cell wall instead. So plasma membranes are a feature of eukaryotic cells, not prokaryotes. This might be a red herring, but I'm not sure if it's the correct answer.

(D) Energy molecules: Both prokaryotic and eukaryotic cells use energy, but the mechanisms are different. Eukaryotes use cellular respiration, while prokaryotes use a process called cellular respiration in the mitochondria. However, this is more about the processes they use rather than a distinguishing feature.

Wait, but the question is about a feature used to distinguish them. So maybe it's about the presence of a cell membrane or a cell wall. Since prokaryotes don't have a cell membrane, but eukaryotes do, that would make (C) plasma membranes incorrect because prokaryotes don't have them. So maybe (B) size differences is the answer because it's a clear and commonly used feature.

Alternatively, I remember that sometimes people confuse the presence of a cell wall with the use of a cell membrane. But the question is about a feature used to distinguish, so the presence of a cell wall (plasma membrane) is a key feature. But wait, prokaryotes don't have a cell membrane, so maybe the answer is (B) size differences.

Wait, but the question is about a feature, not a process. So perhaps the answer is (B) size differences because that's a clear and direct way to tell them apart. Alternatively, (C) plasma membranes might be a distractor because prokaryotes don't have them, but eukaryotes do. So maybe the answer is (B) size differences.

I'm a bit confused now. Let me think again. The question is about a feature used to distinguish prokaryotic from eukaryotic cells. The key feature is the presence of a cell membrane. Since prokaryotes don't have a cell membrane, but eukaryotes do, the presence of a cell membrane is a distinguishing feature. But the options don't include "cell membrane," so maybe the answer is (B) size differences because it's a clear and direct feature.

Alternatively, maybe the answer is (C) plasma membranes because that's a feature of eukaryotes, but the question is about prokaryotes, so that's not it. So I think *the answer is (B) size differences* because that's a clear and direct way to distinguish them. </think>

The correct answer is *(B) size differences.* Prokaryotic cells are generally smaller than eukaryotic cells, and this size difference is a clear feature used to distinguish them. While other features like the presence of a cell membrane or energy molecules are important, size differences are the most straightforward and commonly used feature for this classification.

**Response by retrieval-perturbed model**: <think> Okay, so I need to figure out which feature distinguishes prokaryotic cells from eukaryotic cells. I remember that prokaryotic cells, which are bacteria, have a simpler structure compared to eukaryotic cells, like animals or plants. Let me think about what makes them different. I recall that prokaryotic cells don't have a membrane-bound nucleus. They have a single, circular chromosome. Eukaryotic cells, on the other hand, have a nucleus with membrane-bound organelles. So, maybe the difference is in the structure of the cell itself.

Looking at the options: (A) life processes, (B) size differences, (C) plasma membranes, (D) energy molecules. I think plasma membranes are definitely a key feature. Prokaryotic cells have their own membrane, separate from the bacterial cell wall, while eukaryotic cells have a thick membrane around the nucleus. So the plasma membrane must be the distinguishing feature.

I don't think size differences (B) are the main thing because both prokaryotic and eukaryotic cells can be large. Life processes (A) might refer to functions like growth, reproduction, etc., but that's more about what they do rather than their structure. Energy molecules (D) could be a factor, but the main structural difference is the presence of a nuclear envelope in eukaryotes and the absence in prokaryotes.

So, I'm pretty sure the *correct answer is (C) plasma membranes*. </think>

The correct answer is *(C) plasma membranes*. Prokaryotic cells have their own nuclear envelope and membrane, separate from the bacterial cell wall, while eukaryotic cells have a thick nuclear membrane surrounding their nucleus.

## C.3   AUC Results for More Models and Datasets

Figure 9 displays AUC results from probe analysis for the Phi4-mini-reasoning and R1-Qwen-7B models on the Math&Logic and SciEng fields of the MMLU dataset. The figures demonstrate that critical attention head locations remain remarkably consistent for each model across different

datasets and all four perturbation experiments. This stability strengthens our conclusion that a localized set of neurons governs reasoning-dominant or retrieval-dominant behavior.

Moreover, the specific locations of these critical layers vary across different architectures. Critical layers concentrate in layers 12 through 16 for R1-Llama-8B, layers 15 through 20 for R1-Qwen-7B, and layers 12 through 18 for Phi4-mini-reasoning. These variations indicate that the specific control neuron placement depends on the model architecture.

## C.4    LOSS DYNAMICS

Figure 10 illustrates the training dynamics of FARL by plotting the validation accuracy, reward, and the unlearning loss over time. The steady increase in both validation accuracy and reward confirms that the RL objective is effectively optimized throughout training. Concurrently, the consistent decrease in the NPO loss indicates that our unlearning objective is also successfully met, reflecting that the model is progressively forgetting the targeted retrieval shortcuts as intended.

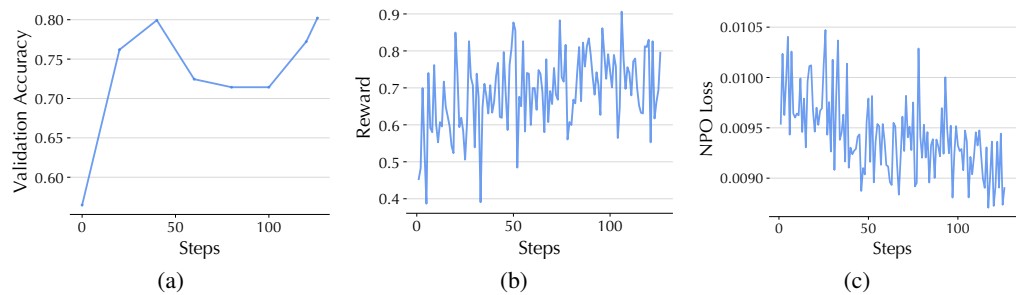

| (a) | (b) | (c) |

Figure 10: The (a) accuracy on the validation dataset, (b) reward score, and (c) NPO loss during the FARL training on R1-Llama-8B.

## C.5    ADDITIONAL FARL RESULTS ON R1-QWEN-7B

Table 2: Training and reasoning performance comparison between the proposed and comparison methods.

| Method | Perturbation Metric | | Performance Metric (Training Domain) | | Performance Metric (Out of Domain) | | Training Time |
|---|---|---|---|---|---|---|---|
| | R-PSR ↓ | T-PSR ↓ | MTL | ACC ↑ | MTL | ACC ↑ | |
| R1-Qwen-7B (Base) | 0.762 | 0.059 | 1244.0 | 0.769 | 1237.6 | 0.690 | / |
| SFT | 0.735 | 0.073 | 1177.8 | 0.758 | 1412.2 | 0.687 | 9m 36s |
| RL (GRPO) | 0.356 | 0.231 | 1481.9 | 0.932 | 1486.4 | 0.712 | 4h 09m 35s |
| FARL | 0.295 | 0.201 | 1660.8 | 0.924 | 1633.5 | 0.724 | 4h 22m 53s |

Table 2 presents the additional results of FARL and comparison methods on R1-Qwen-7B. The results demonstrate that FARL successfully reduces the overall perturbation rate by 60.4%, thereby outperforming all comparison methods. This reduction indicates that FARL effectively makes the model's reasoning mechanism dominant while enhancing the robustness of the CoT.

Regarding performance metrics, the SFT-trained model exhibits an accuracy drop of 1.4% within the mathematical training domain and a minimal 0.4% decrease beyond the training domain. In contrast, RL achieves substantial improvements of 21.3% and 3.2% within and outside the training domain, respectively. These findings reinforce our conclusions that RL more effectively enhances the model's reasoning ability and demonstrates superior generalization compared to SFT.

Moreover, FARL achieves the strongest accuracy improvements both in-domain (20.1%) and out-of-domain (5.0%). According to Figure 11, FARL also boosts the cycle, diameter, and small world index by 14.9%, 1.97%, and 15.1%, respectively, when compared with the base model, surpassing all comparison methods. These findings provide additional support for our design intuition that suppressing the retrieval mechanism during RL can further elicit the model's reasoning ability across accuracy, generalization, and quality dimensions.

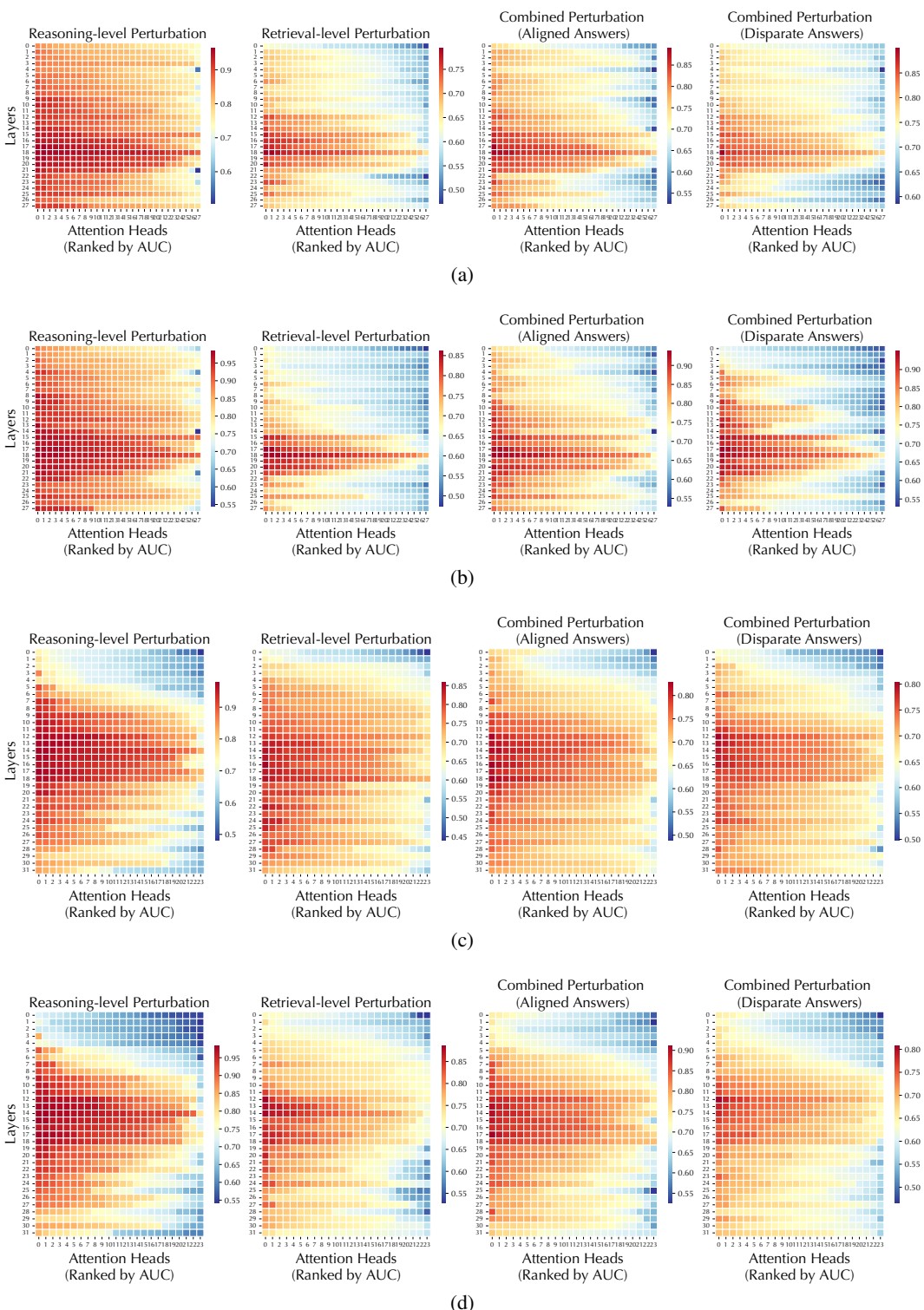

Figure 9: AUC results of (a) R1-Qwen-7B on Math&Logic field, (b) R1-Qwen-7B on SciEng field, (c) Phi4-mini-reasoning on Math&Logic field, and (d) Phi4-mini-reasoning on SciEng field of MMLU dataset in the experiments of reasoning-level perturbation, retrieval-level perturbation, combined perturbation with the same target answer, and with disjoint target answers.

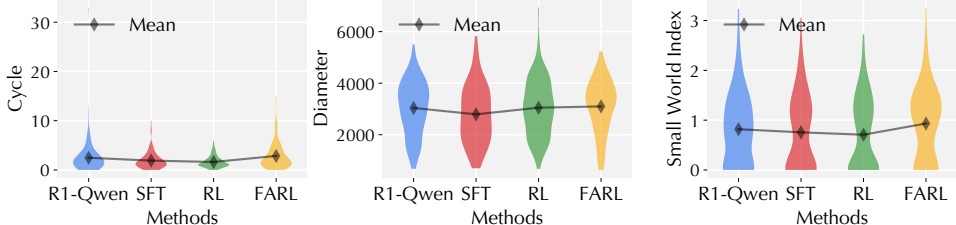

Figure 11: Cycle, diameter, and small world index distributions of the reasoning graph generated by LRMs trained with FARL and baselines.

# D    ADDITIONAL DISCUSSION

## D.1    RELIABILITY OF FINE-TUNING-BASED MEMORY EDITING

While most existing mechanistic editing methods (Meng et al., 2022; 2023) are designed to modify facts (i.e., subject-relation-object triples) stored in LLMs, in this work, we need to edit general question-answer mappings. This distinction renders mechanistic editing methods less applicable. We therefore adopt supervised fine-tuning (SFT), a general yet effective method for modifying internal knowledge that enables us to test whether edited knowledge can be successfully retrieved at inference time. This choice is well-justified: recent work by Gangadhar & Stratos (2024) and Meng et al. (2022) shows that SFT achieves competitive performance with specialized mechanistic editing methods such as ROME (Meng et al., 2022) and MEMIT (Meng et al., 2023) on standard knowledge editing benchmarks, establishing it as a rigorous baseline for our investigation.

To empirically validate the reliability of SFT-based editing, following prior work, we measure its performance using three key metrics (Gangadhar & Stratos, 2024):

**Efficacy.** Let $x$ denote a question and $y^*$ be its target answer. Efficacy measures the probability that model $\mathcal{M}$ produces $y^*$ as the answer to $x$: $y^* = \arg\max_y \mathcal{M}_\theta(y|x)$. To isolate the effect of memory from reasoning capability, we directly extract the answer using the prompt "The correct answer is (" without CoT. As shown in Table 3, the majority of samples across all domains are successfully perturbed, indicating that SFT effectively alters the LRM's memory.

Table 3: Efficacy of SFT-based editing across domains.

| Domain | MathLogic | SciEng | Computing | LifeSci | Health | BusinessEcon |
|---|---|---|---|---|---|---|
| Efficacy | 0.975 | 0.925 | 0.9 | 0.925 | 0.95 | 0.875 |

| Domain | Society | Humanities | Arc_Easy | Arc_Challenge | GPQA |
|---|---|---|---|---|---|
| Efficacy | 0.9 | 0.925 | 0.950 | 0.9 | 0.925 |

**Generalization.** We further examine the robustness of the editing to question variation by measuring generalization: the probability that model $\mathcal{M}$ produces $y^*$ as the answer to a paraphrased question $\tilde{x}$: $y^* = \arg\max_y \mathcal{M}_\theta(y|\tilde{x})$. To this end, we randomly sample 50 successfully perturbed examples from each domain in the perturbation-to-retrieval experiment and generate paraphrased questions using GPT-5-mini. As shown in Table 4, the generalization measures remain high across domains, indicating that the editing exhibits strong robustness to question paraphrasing.

Table 4: Generalization of SFT-based editing across domains.

| Domain | MathLogic | SciEng | Computing | LifeSci | Health | BusinessEcon |
|---|---|---|---|---|---|---|
| Generalization | 0.975 | 0.925 | 0.9 | 0.925 | 0.95 | 0.875 |

| Domain | Society | Humanities | Arc_Easy | Arc_Challenge | GPQA |
|---|---|---|---|---|---|
| Generalization | 0.9 | 0.925 | 0.950 | 0.9 | 0.925 |

**Locality.** We also assess whether SFT-based editing inadvertently affects unrelated knowledge by measuring locality. Formally, locality quantifies the probability that $\mathcal{M}$ generates the ground-truth answer $y'$ for an unrelated question $x'$: $y' = \arg\max_y \mathcal{M}_\theta(y|x')$. To evaluate this, we fine-tune the model exclusively on the Math&Logic domain using incorrect answers and measure performance on all samples in other domains both before and after SFT. Table 5 compares the locality before and after SFT under two settings: CoT enabled and CoT disabled. In both cases, the locality after SFT is almost identical to that before SFT, suggesting that the intervention produces targeted, localized changes rather than inducing global knowledge degradation.

Table 5: Locality of SFT-based editing across domains.

|  | Before SFT | After SFT |
|---|---|---|
| w/o CoT | 0.261 | 0.263 |
| w/ CoT | 0.713 | 0.716 |

## D.2 FREE-FORM QUESTION ANSWERING

To validate the generalizability of our conclusion to non-multiple-choice QA, we randomly sample 400 questions from the free-form QA dataset GeneralThought (GeneralReasoning, 2025), which span diverse question types and domains and have answers that are relatively easy to verify (e.g., numbers, nouns, short phrases). We then use GPT-5-mini to produce the misleading answers $y_r$ and $y_t$ and run the perturbation pipeline introduced in §3.1. Table 6 reports the joint influence of reasoning and retrieval perturbations, showing that their effects closely match those observed on the multiple-choice QA dataset (e.g., Figure 2).

Table 6: Joint influence of retrieval and reasoning on free-form QA.

| Model  Metric | T-PSR | R-PSR | T-PSR + R-PSR (Aligned Answers) | R-PSR (Disparate Answers) | T-PSR (Disparate Answers) |
|---|---|---|---|---|---|
| R1-Llama-8B | 0.320 | 0.563 | 0.652 | 0.458 | 0.181 |
| R1-Qwen-7B | 0.292 | 0.642 | 0.772 | 0.121 | 0.552 |
| Phi4-mini | 0.095 | 0.352 | 0.431 | 0.345 | 0.185 |
| Qwen3-8B | 0.145 | 0.179 | 0.340 | 0.277 | 0.093 |

## D.3 POSITIONAL BIAS IN REASONING PERTURBATION

Notably, recency bias occurs naturally even without perturbation: for instance, the conclusions of reasoning typically appear at the end of CoTs. Since our goal is to evaluate the relative dominance of reasoning versus retrieval under different perturbations, recency bias does not confound our findings. Under the combined perturbation (Figure 3), when reasoning-led answer $y_r$ and retrieval-led answer $y_t$ conflict, the retrieval capability often overrides $y_r$, showing inherent competition between reasoning and retrieval pathways rather than positional bias.

To empirically test the impact of hint positioning, we vary the hint's position between the beginning and end of the CoT and report the corresponding perturbation success rate (R-PSR) in Table 7. Observe that R-PSR appears insensitive to the hint's position, indicating the limited impact of positional bias.

In sum, our perturbation design for CoT is simple yet effective, enabling causal sensitivity tests of how the final answer depends on the reasoning mechanism. Furthermore, the combined perturbation (§3.3) and ablation analysis (§3.4) provide direct evidence that both reasoning and retrieval influence the final answer far more than positional bias.

Table 7: R-PSR comparison between different hint positions.

| Model\Insertion Position | Beginning of CoT | End of CoT |
|---|---|---|
| R1-Llama-8B | 0.602 | 0.665 |
| R1-Qwen-7B | 0.634 | 0.698 |
| Phi4-mini | 0.431 | 0.441 |
| Qwen3-8B | 0.233 | 0.253 |

# E LLM USAGE STATEMENT

We employed large language models solely for language refinement and polishing. Importantly, this research does not rely on LLMs for any substantive, original, or non-standard components.

