# OpenReview forum: "Reasoning or Retrieval? A Study of Answer Attribution on Large Reasoning Models"
_ICLR.cc/2026/Conference — ICLR 2026 Poster_

### Official Review · Reviewer_3Cxz · 2025-10-28

**Soundness:** 2
**Presentation:** 3
**Contribution:** 2
**Rating:** 4
**Confidence:** 3

**Summary:**

This paper make the mechanism study towards the chain-of-thought process of Large Reasoning Models (LRM). They conduct controlled experiments to analyse whether LRM generate answer by CoT reasoning or memory retrieval.  Based on that, the paper answer 3 research questions: RQ1: Do LRMs employ reasoning and retrieval simultaneously to derive answers? RQ2: What factors influence the dominance of one capability over the other? RQ3: How can we control the relative strength of these capabilities? Build upon those findings, the author propose a novel fine-tuning framework that integrates memory unlearning with reinforcement learning that enhances generalizable reasoning capabilities.

**Strengths:**

The experiment is conducted among different types of model size, architecture, and training paradigm, which make to conclusion relative plausible. The paper employs machine unlearning method to optimize training process, which enhances generalizable reasoning capabilities.

**Weaknesses:**

Though the experiment is conducted thoroughly, the memory and reasoning perturbation mechanism is not plausible enough and needs further discussion, see question part for detals.

**Questions:**

1. reasoning perturbation is achieved by placing error answer in the end of thinking phase. However, the design seems to be simple or naive, since transformer mechanism normally focus on recent tokens (and attention sink), which could make LRM ignore the previous thinking process and make shortcuts.
2. memory perturbation is fine-tuning qa-pairs to change memories of LRM, though the author claims that fine-tuned phase is restricted to relevant knowledge and minimize side-effect, I am not sure how this can be guaranteed. For example, deteriorate the reasoning capability of LRM, which might in-turn weakening the experiment conclusion. More ablations would by beneficial.
3. the paper disentangle reasoning and memory by whether letting LRM to think or not, though it is simple and effective, however, during reasoning phase, some knowledge could be retrieved from LRM's memories.
4. FARL experiment lacks of discussion of the training dataset.

If those concerns are settled properly, I am willing to raise my score.

---

> ### Author Response · Authors · 2025-11-20
>
> We thank the reviewer for the valuable feedback on improving this paper! Please find below our response to the reviewer’s questions.
>
> ---
>
> > reasoning perturbation is achieved by placing error answer in the end of thinking phase. However, the design seems to be simple or naive, since transformer mechanism normally focus on recent tokens (and attention sink), which could make LRM ignore the previous thinking process and make shortcuts.
>
> We appreciate the insightful comments. Notably, recency bias occurs naturally even without perturbation: for instance, the conclusions of reasoning typically appear at the end of CoTs. Since our goal is to evaluate the relative dominance of reasoning versus retrieval under different perturbations, recency bias does not confound our findings. Under the combined perturbation (Figure 2), when reasoning-led answer $y_r$​ and retrieval-led answer $y_t$​ conflict, the retrieval capability often overrides $y_r$​, showing inherent competition between reasoning and retrieval pathways rather than positional bias.
>
> To empirically test the impact of hint positioning, we vary the hint's position between the beginning and end of the CoT and report the corresponding perturbation success rate (R-PSR) in the table below. Notably, R-PSR appears insensitive to the hint's position, indicating the limited impact of positional bias.
>
> In sum, our perturbation design for CoT is simple yet effective, enabling causal sensitivity tests of how the final answer depends on the reasoning mechanism. Furthermore, the combined perturbation (Section 3.3) and ablation analysis (Section 3.4) provide direct evidence that both reasoning and retrieval influence the final answer far more than positional bias.
>
> | Model\Insertion Position| Beginning of CoT | End of CoT |
> |--------------------|:-------------------:|:--------------:|
> | R1-Llama-8B        |        0.602        |      0.665     |
> | R1-Qwen-7B         |        0.634        |      0.698     |
> | Phi4-mini          |        0.431        |      0.441     |
> | Qwen3-8B           |        0.233        |      0.253     |
>
> ---
>
> > memory perturbation is fine-tuning qa-pairs to change memories of LRM, though the author claims that fine-tuned phase is restricted to relevant knowledge and minimize side-effect, I am not sure how this can be guaranteed. For example, deteriorate the reasoning capability of LRM, which might in-turn weakening the experiment conclusion. More ablations would by beneficial.
>
> To directly measure whether memory perturbation (SFT) inadvertently affects the model’s reasoning capability, we conduct additional experiments using the metric of locality [1] (more details in Appendix D.1 of the revised paper). Intuitively, locality quantifies the probability that model $\mathcal{M}$, under perturbation to target question $x$, generates the ground-truth answer $y'$ for an unrelated question $x'$: $y' = \arg\max_y\mathcal{M}_\theta( y|x')$. To evaluate this, we fine-tune the model exclusively on the Math&Logic domain using incorrect answers and measure performance on all samples in other domains both before and after SFT. The table below compares the locality before and after SFT under two settings: CoT enabled and CoT disabled. In both cases, the locality after SFT is almost identical to that before SFT, suggesting that the intervention produces targeted, localized changes rather than inducing global knowledge/capability degradation.
>
> |  | Before SFT | After SFT |
> |:-----------:|:-----------:|:-----------:|
> | w/o CoT | 0.261 | 0.263 |
> | w/ CoT | 0.713 | 0.716 |
>
> ---
>
> > the paper disentangle reasoning and memory by whether letting LRM to think or not, though it is simple and effective, however, during the reasoning phase, some knowledge could be retrieved from LRM's memories.
>
> We thank the reviewer for this insightful comment. Indeed, LRMs must retrieve foundational knowledge during reasoning (e.g., mathematical axioms). However, our focus is on disentangling the source of the final answer: whether it derives from explicit reasoning chains or direct question-answer mappings. This distinction is critical for addressing CoT-answer inconsistencies and improving RL effectiveness.
>
> ---
>
> > FARL experiment lacks of discussion of the training dataset.
>
> We thank the reviewer for pointing this out. We used the Math&Logic domain of the MMLU dataset for both SFT and RL. This choice was informed by our findings in Figure 2, which demonstrate this domain's reasoning-intensive nature. The dataset comprises 1,353 training samples and 147 validation samples. A detailed composition of this domain is provided in Appendix B1. We have added this clarification in the revised paper.
>
> ---
>
> [1] Gangadhar et al., Model Editing by Standard Fine-Tuning, ACL’24.
>
> ---
>
> Again, we thank the reviewer for the valuable feedback. Please let us know if there are any other questions or suggestions.
>
> Best,
>
> Authors

---

### Official Review · Reviewer_kodm · 2025-10-29

**Soundness:** 3
**Presentation:** 3
**Contribution:** 3
**Rating:** 6
**Confidence:** 4

**Summary:**

Given the phenomenon of LRMs that reasoning traces often contradict with final answers, this paper investigates what factor influences the reasoning ability, in terms of the conflict between retrieval of LRMs' knowledge (internal knowledge) and Chain-of-Thought (external prompting).

1. The authors find that both COT and internal knowledge contribute to the final answers.
2. The authors investigate how some competing factors(model sizes, reasoning problem domains, reasoning model training methods) influence the reasoning abilities of LLMs. The authors also validate the "post-hoc exaplanation phenomenon" (models try to fabricate reasoning steps to derive a false answer), further prove that reasoning ability that distill from LRMs is not reliable as RL.
3. To validate the findings above, the authors suggest apply RL on reasoning-intensive datasets to avoid the model from retrievaling results from their own knowledge. The authors propose to train model with knowledege unlearning method to forget specific knowledge(with GRPO and NPO specifically).

**Strengths:**

1. The authors conduct sufficient empirical experiments to validate how the factors of COT prompts and internal model knowledge influence the final results.
2. The authors conduct "Post-Hoc Explanation" experiments to cross validate the results of previous methods.
3. The authors leverage an unlearning methods, which add NPO after the GRPO to demonstrate that weaken the knowledge ability of LRMs would enhance the reasoning ability of LLMs, in terms of reasoning robustness and effectiveness. The authors performance extra evaluation on reasoning path quality.

**Weaknesses:**

1. The knowledge and reasoning ability may still hard to decouple to analysis. The SFT attack may still harm the reasoning ability of COT,As experiments in table1 demonstrates, SFT give worse R-PSR than original R1-Llama-8B. The authors just claim the assumetion in Line 150 that the impact would be small.
2. Some claims seem not proper:
    * In line 48, this research seems still not "a mechanistic understanding of how different capabilities jointly influence LRMs’ answer generation"
    * From line 360 to line 362,It is not easy to understand that "our findings reveal a challenge where the retrieval mechanism enables models to “hack” the reward signal during RL and impair its effectiveness". As you will find if you estimate $\delta=T-PSR - PER$, which accounts for LRMs own reasoning failure, it is even more than PER.
3. Current study mainly focus on Multi-choice QA, but more open-ended problems could be studied in the future.

**Questions:**

1. The section `Attention Patterns` seems not related to other sections, especially the results is not used for RQ3. What's the purpose of this section.
2. Some results settings:
    * How did the authors get the result of Fig2 on different datasets?
    * What's the experiment settings of RQ3 Table1 to get the performance? As previous "pertubation attack" only conduct on the questions that LLM can correctly solve.
3. The authors aim to interval the "retrieval knowledge shortcut" to enhance the reasoning ability of models. However, the actual reasoning accuracy may decrease. As the model still need the corrpsond basic knowledge to perform get the results. What if the authors test the result of unlearning some intermediate knowledge used by COT, for those $y_r$ or $y$ results?
4. As the reasoning ability could be enhanced when LLMs are unlearned the final knowledge, if the LLMs can access external knowledge, can they achieve better accuracy on harder problem?

---

> ### Author Response · Authors · 2025-11-20
>
> We thank the reviewer for the valuable feedback on improving this paper! Please find below our response to the reviewer’s questions.
>
> ---
>
> > The knowledge and reasoning ability may still hard to decouple to analysis. The SFT attack may still harm the reasoning ability of COT, As experiments in table1 demonstrates, SFT give worse R-PSR than original R1-Llama-8B. The authors just claim the assumetion in Line 150 that the impact would be small.
>
> We appreciate the reviewer's comments and address them as follows:
>
> - First, the slight difference in R-PSR between base and SFT models in Table 1 can be attributed to the sample selection: when measuring R-PSR, we restrict perturbations to samples that each model initially answers correctly. Since the SFT model achieves higher accuracy than the base model (e.g., 0.787 vs. 0.725), a larger set of samples is perturbed when measuring the SFT model's R-PSR.
>
> - Further, to directly measure whether SFT inadvertently affects the model’s reasoning capability, we conduct additional experiments using the metric of locality [1] (more details in Appendix D.1 of the revised paper). Intuitively, locality quantifies the probability that model $\mathcal{M}$, under perturbation to target question $x$, generates the ground-truth answer $y'$ for an unrelated question $x'$: $y' = \arg\max_y\mathcal{M}_\theta( y|x')$. To evaluate this, we fine-tune the model exclusively on the Math&Logic domain using incorrect answers and measure performance on all samples in other domains both before and after SFT. The table below compares the locality before and after SFT under two settings: CoT enabled and CoT disabled. In both cases, the locality after SFT is almost identical to that before SFT, suggesting that the intervention produces targeted, localized changes rather than inducing global knowledge/capability degradation.
>
> |  | Before SFT | After SFT |
> |:-----------:|:-----------:|:-----------:|
> | w/o CoT | 0.261 | 0.263 |
> | w/ CoT | 0.713 | 0.716 |
>
> - Finally, our analyses in RQ1 and RQ2 examine the existence of competing mechanisms in answer generation and the factors governing their relative dominance. While completely isolating reasoning from retrieval capabilities may be impossible, our conclusions remain robust because they derive from relative comparisons across domains and models rather than absolute measurements.
>
> ---
>
> [1] Gangadhar et al., Model Editing by Standard Fine-Tuning, ACL’24.

---

> ### Author Response · Authors · 2025-11-20
>
> > Some claims seem not proper: 1) In line 48, this research seems still not "a mechanistic understanding of how different capabilities jointly influence LRMs’ answer generation". 2) From line 360 to line 362,It is not easy to understand that "our findings reveal a challenge where the retrieval mechanism enables models to “hack” the reward signal during RL and impair its effectiveness". As you will find if you estimate \sigma = T - PSR - PER , which accounts for LRMs own reasoning failure, it is even more than PER. 3) Current study mainly focus on Multi-choice QA, but more open-ended problems could be studied in the future
>
> We appreciate the reviewer’s comments.
>
> 1) We have revised the claim as “we still lack an understanding of how different capabilities jointly influence LRMs' answer generation and what factors determine their relative dominance” to better reflect this work’s focus. Further, we have conducted additional experiments to study the dynamics of retrieval-reasoning interactions at each reasoning step through the logit lens (Appendix D.2).
>
> 2) We have revised the claim as “our findings reveal a potential risk: the retrieval mechanism may interfere with the reasoning mechanism, enabling models to take shortcuts during RL and undermining its effectiveness” for clarity.  To clarify, PER represents a subset of T-PSR, corresponding to cases where the model fabricates CoTs to justify retrieved incorrect answers, a behavior we refer to as post-hoc explanation. The difference (T-PSR − PER) corresponds to cases where CoTs are inconsistent with retrieved answers. Different models exhibit varying PER vs. (T-PSR − PER) ratios; for instance, distilled models tend to have higher PER values.
>
> 3) To validate the generalizability of our conclusion to non-multiple-choice QA, we use GeneralThought [2], a free-form QA dataset that spans diverse question types and domains (more details in Appendix D.4 of the revised paper). We randomly sample 400 questions with answers that are relatively easy to verify (e.g., numbers, nouns, short phrases). We then use GPT-5-mini to produce the misleading answers $y_r$ and $y_t$ and run the perturbation pipeline introduced in Section 3.1. The table below reports the joint influence of reasoning and retrieval perturbations, showing that their effects closely match those observed on the multiple-choice QA dataset (e.g., Figure 2).
>
> | Model \ Metric | T-PSR | R-PSR | T-PSR + R-PSR (Aligned Answers) | R-PSR (Disparate Answers) | T-PSR (Disparate Answers) |
> |----------------|:-----:|:-----:|:-------------------------------:|:-------------------------:|:-------------------------:|
> | R1-Llama-8B    | 0.320 | 0.563 |              0.652              |           0.458           |           0.181           |
> | R1-Qwen-7B     | 0.292 | 0.642 |              0.772              |           0.121           |           0.552           |
> | Phi4-mini      | 0.095 | 0.352 |              0.431              |           0.345           |           0.185           |
> | Qwen3-8B       | 0.145 | 0.179 |              0.340              |           0.277           |           0.093           |
>
> ---
>
> > The section Attention Patterns seems not related to other sections, especially the results is not used for RQ3. What's the purpose of this section.
>
> In RQ2, we analyze key factors that may impact the relative dominance of reasoning and retrieval capabilities, including both external factors (e.g., domain and model size) and internal factors (examined through attention pattern analysis). In this analysis, we identify specific attention heads that may govern the relative dominance of the two mechanisms.
>
> To verify that the identified attention heads can causally control the selection between the two pathways, we conduct additional experiments based on activation patching [3] (more details in Appendix D.3 of the revised paper).
>
> Under the perturbation-to-retrieval setting, we collect attention activations from (i) successfully perturbed samples in the SFT-tuned model (perturbation run) and (ii) their corresponding activations in the base model (clean run). We then conduct two complementary interventions:
>
> - Replacing perturbation-run activations with clean-run activations for the attention heads ranked in the top 5% by AUC score (Figure 5) restores the original answers with 87.2% success rate. The same intervention on an equal number of randomly selected attention heads yields only 5.3% restoration.
>
> - The reverse intervention, patching clean-run activations with perturbation-run activations, produces perturbed answers with 89.5% success rate for the top 5% AUC heads, but only 2.1% for randomly selected heads.
>
> These results indicate that high-AUC attention heads exert causal control over the competition between reasoning and retrieval pathways, with the final answer exhibiting clear counterfactual dependence on their activation patterns.

---

> ### Author Response · Authors · 2025-11-20
>
> > Some results settings: 1) How did the authors get the result of Fig2 on different datasets? 2) What's the experiment settings of RQ3 Table1 to get the performance? As previous "pertubation attack" only conduct on the questions that LLM can correctly solve.
> We appreciate the reviewer for raising these questions.
>
> 1) Figure 2 contains results from four types of experiments: (i) perturbation to reasoning, (ii) perturbation to retrieval, (iii) combined perturbation with aligned answers, and (iv) combined perturbation with divergent answers. The four perturbation measurements in each group correspond to these settings from left to right. The fourth bar is shown with two colors because some predictions follow the reasoning-perturbation answer $y_r$​ (R-PSR) while others follow the retrieval-perturbation answer $y_t$​ (T-PSR). All experiments are conducted separately on different datasets and domains. We then aggregate the corresponding metrics to produce the results shown in Figure 2.
>
> 2) Yes, we filter for initially correct answers separately for each experimental setting, including those reported in Table 2. Although the total number of samples differs across experiments, our conclusions remain valid because all comparisons are based on relative proportions rather than absolute counts.
>
> ---
>
> > The authors aim to interval the "retrieval knowledge shortcut" to enhance the reasoning ability of models. However, the actual reasoning accuracy may decrease. As the model still need the corrpsond basic knowledge to perform get the results. What if the authors test the result of unlearning some intermediate knowledge used by COT, for those y_r or y results.
>
> We would like to clarify two key points: (i) FARL unlearns direct question-answer mappings rather than the foundational knowledge required for deriving individual reasoning steps; (ii) removing the retrieval shortcut alone does not improve reasoning ability. In FARL, RL is the primary driver of reasoning improvement, while unlearning suppresses the retrieval shortcut to enhance RL's effectiveness. Therefore, unlearning intermediate knowledge in isolation would only degrade reasoning ability, leaving the model's reliance on retrieval for final answers intact.
>
> To verify this, we conduct an additional experiment unlearning intermediate reasoning steps. We collect clean reasoning traces from the Computing domain, split each CoT into steps (delimited by "\n\n"), and randomly unlearn 50% of the steps. This causes accuracy to drop sharply from 0.708 to 0.324.
>
> ---
>
> > As the reasoning ability could be enhanced when LLMs are unlearned the final knowledge, if the LLMs can access external knowledge, can they achieve better accuracy on harder problem?
>
> We appreciate this interesting question. We evaluated the baselines and FARL on the AIME 2025 dataset, which contains 30 challenging math problems. To provide the models with external knowledge, we included five in-context exemplars drawn from AIME 2024 problems with their solutions. As shown in the table below, FARL achieves the highest accuracy among all models, indicating that unlearning question-answer mappings while providing external knowledge may improve performance on harder problems. However, the absolute accuracy remains modest, suggesting that performance may be largely constrained by the base model's reasoning capability and the quality of the retrieved exemplars.
>
> |        | Base (R1-Llama-8B) | SFT |  RL | FARL |
> |:------:|:------------------:|:---:|:---:|:----:|
> | Pass@1 |         0.20        | 0.20 | 0.20 | 0.23 |
>
> ---
>
> [2] GeneralThought dataset: huggingface.co/datasets/RJT1990/GeneralThoughtArchive\
> [3] Meng et al., Locating and Editing Factual Associations in GPT, NeurIPS’22.\
> [4] Ortu et al., Competition of mechanisms: Tracing how language models handle facts and counterfactuals, ACL’24.
>
> ---
>
> Again, we thank the reviewer for the valuable feedback. Please let us know if there are any other questions or suggestions.
>
> Best,
>
> Authors

---

> > ### Comment · Reviewer_kodm · 2025-11-26
> >
> > Thanks for the responses. I will maintain my current positive rating.

---

### Official Review · Reviewer_fSyB · 2025-10-31

**Soundness:** 2
**Presentation:** 3
**Contribution:** 2
**Rating:** 2
**Confidence:** 3

**Summary:**

Large Reasoning Models (LRMs) generate answers using two competing mechanisms a) Chain-of-Thought (CoT) Reasoning and b) Memory Retrieval. These mechanisms can conflict, leading to inconsistencies between reasoning traces and final answers. To show that LRMs LLM use both reasoning and retrieval simultaneously, authors designed controlled perturbation experiments to perturb either reasoning or retrieval. The reasoning steps are subtly altered to be misleading or incorrect. Or, the model’s memory is poisoned with misleading cues. They find that smaller or distilled models are more vulnerable to retrieval perturbations. They might fabricate reasoning traces to justify retrieved answers. In comparison, larger models and those trained with reinforcement learning are more robust and reasoning-driven. Given on this experiment, authors proposed FARL (Forgetting-Augmented Reinforcement Learning) to: a) suppress retrieval shortcuts, b) enhance reasoning-dominant behavior, c) improve generalization and robustness. It achieves 47.8% improvement in CoT robustness, 22.8% accuracy gain in-domain tasks, and 5.8% accuracy gain out-of-domain tasks.

**Strengths:**

The methodology is rigorous, including controlled perturbation experiments and attention head analysis, which strengthen the validity of the findings.

Understanding the interplay between reasoning and retrieval is critical for advancing trustworthy AI, especially in high-stakes domains like math, logic, and scientific reasoning.

**Weaknesses:**

The paper primarily focuses on math and logic tasks to evaluate reasoning vs. retrieval. They are limited on multiple-choice QA, may not generalize to open source qa. Please add an evaluation on free-form answers with verifiable graders, such as [GeneralThought](https://huggingface.co/datasets/RJT1990/GeneralThoughtArchive).

R-PSR and T-PSR  are correlational indicators, not causal evidence of pathway dominance. A misleading cue that flips an answer does not prove the answer was reasoning-driven. Can we add causal intervention experiments (e.g., targeted weight/activation ablations on putative retrieval heads; causal scrubbing on residual streams) to show counterfactual dependence of the final answer on each pathway?

Metrics like R-PSR and T-PSR are binary and may not capture nuanced interactions between reasoning and retrieval. For example, a model might partially rely on both mechanisms in a non-exclusive way. What if we can add token-level attributions such as logit lens analyses over steps?

The extraction of answers and judgment of reasoning correctness occasionally falls back to GPT-4o-mini for answer extraction (section 3.2). Does misclassification inflate perturbation success? I suggest that authors add a human-validated subset or at least a majority-vote ensemble judge.

**Questions:**

How does NPO work in FARL? Did you input all x and y to the NPO?  How does it compell models to “forget” specific memorized answers?

Unify b_x (line 4) and x (line 9) in Algorithm 1 for easier to understand

---

> ### Author Response · Authors · 2025-11-20
>
> We thank the reviewer for the valuable feedback on improving this paper! Please find below our response to the reviewer’s questions.
>
> ---
>
> > The paper primarily focuses on math and logic tasks to evaluate reasoning vs. retrieval. They are limited on multiple-choice QA, may not generalize to open source qa. Please add an evaluation on free-form answers with verifiable graders, such as GeneralThought.
>
> Following the reviewer’s suggestion, we use GeneralThought, a free-form QA dataset that spans diverse question types and domains (more details in Appendix D.4 of the revised paper). We randomly sample 400 questions with answers that are relatively easy to verify (e.g., numbers, nouns, short phrases). We then use GPT-5-mini to produce the misleading answers $y_r$ and $y_t$ and run the perturbation pipeline introduced in Section 3.1. The table below reports the joint influence of reasoning and retrieval perturbations, showing that their effects closely match those observed on the multiple-choice QA dataset (e.g., Figure 2).
>
> | Model \ Metric | T-PSR | R-PSR | T-PSR + R-PSR (Aligned Answers) | R-PSR (Disparate Answers) | T-PSR (Disparate Answers) |
> |----------------|:-----:|:-----:|:-------------------------------:|:-------------------------:|:-------------------------:|
> | R1-Llama-8B    | 0.320 | 0.563 |              0.652              |           0.458           |           0.181           |
> | R1-Qwen-7B     | 0.292 | 0.642 |              0.772              |           0.121           |           0.552           |
> | Phi4-mini      | 0.095 | 0.352 |              0.431              |           0.345           |           0.185           |
> | Qwen3-8B       | 0.145 | 0.179 |              0.340              |           0.277           |           0.093           |
>
> ---
>
> > R-PSR and T-PSR are correlational indicators, not causal evidence of pathway dominance. A misleading cue that flips an answer does not prove the answer was reasoning-driven. Can we add causal intervention experiments (e.g., targeted weight/activation ablations on putative retrieval heads; causal scrubbing on residual streams) to show counterfactual dependence of the final answer on each pathway?
>
> Following the reviewer’s suggestion, we perform a causal intervention experiment based on activation patching [1] to verify that the attention heads identified in our attention pattern analysis (Section 3.4) causally control the selection between the two pathways. The experiment is conducted on R1-Llama-8B using the MathLogic domain of the MMLU dataset  (more details in Appendix D.3 of the revised paper).
>
> Under the perturbation-to-retrieval setting, we collect attention activations from (i) successfully perturbed samples in the SFT-tuned model (perturbation run) and (ii) their corresponding activations in the base model (clean run). We then conduct two complementary interventions:
>
> - Replacing perturbation-run activations with clean-run activations for the attention heads ranked in the top 5\% by AUC score (Figure 5) restores the original answers with 87.2% success rate. The same intervention on an equal number of randomly selected attention heads yields only 5.3% restoration.
>
> - The reverse intervention, patching clean-run activations with perturbation-run activations, produces perturbed answers with 89.5% success rate for the top 5\% AUC heads, but only 2.1% for randomly selected heads.
>
> These results indicate that high-AUC attention heads exert causal control over the competition between reasoning and retrieval pathways, with the final answer exhibiting clear counterfactual dependence on their activation patterns.
>
> ---
>
> [1] Meng et al., Locating and Editing Factual Associations in GPT, NeurIPS’22

---

> ### Author Response · Authors · 2025-11-20
>
> > Metrics like R-PSR and T-PSR are binary and may not capture nuanced interactions between reasoning and retrieval. For example, a model might partially rely on both mechanisms in a non-exclusive way. What if we can add token-level attributions such as logit lens analyses over steps?
>
> To elucidate the dynamics of retrieval-reasoning interaction at each reasoning step, we track the logits of the retrieval-led answer $y_t$ and the reasoning-led answer $y$ throughout the model's reasoning process (more details in Appendix D.2 of the revised paper). We employ the perturbation-to-retrieval setting (Section 3.1), where the model has been fine-tuned via SFT to associate the question with an incorrect answer $y_t$, while the reasoning leads to the correct answer $y$.
>
> We split the model's generated reasoning into discrete steps (delimited by *"\n\n"*). We then progressively prefill the context with each reasoning step and probe the model's answer distribution at each step using the prompt *``The correct answer is (''*. Figure 11 illustrates a dynamic competition between retrieval and reasoning:
>
> - Reasoning-Dominant Cases: Figures 11(a) and 11(b) present two representative examples where the model converges on the reasoning-led answer $y$. In both cases, the logit for the perturbed answer $y_t$ begins high but steadily declines as reasoning progresses. Conversely, the logit for $y$ starts low but increases monotonically throughout the reasoning chain, eventually suppressing $y_t$.
>
> - Retrieval-Dominant Cases: In contrast, Figures 11(c) and 11(d) show two representative examples where the model produces the retrieval-led answer $y_t$. Here, the logit of $y_t$ remains higher than $y$ throughout the entire reasoning process, indicating that retrieval consistently overrides the reasoning pathway.
>
> These fine-grained logit trajectories provide direct evidence of the dynamic interaction between retrieval and reasoning at each step of the reasoning process. The divergent patterns across examples support our claim that these two pathways tend to compete with each other intricately.
>
> ---
>
> > The extraction of answers and judgment of reasoning correctness occasionally falls back to GPT-4o-mini for answer extraction (section 3.2). Does misclassification inflate perturbation success? I suggest that authors add a human-validated subset or at least a majority-vote ensemble judge.
>
> To validate the reliability of our automated answer extraction pipeline, we manually inspected 100 randomly sampled examples and found 100% accuracy. This high precision stems from two design choices: First, we strictly constrain the model's output format using the prompt suffix “Output format: The correct answer is (A/B/C/D)", which ensures answers are easily extractable. Second, our LLM-as-judge implementation includes a fault-tolerant mechanism to handle edge cases where no valid choice is generated (i.e., a “None” answer is given), thereby preventing extraction errors from propagating.
>
> ---
>
> > How does NPO work in FARL? Did you input all x and y to the NPO? How does it compell models to “forget” specific memorized answers?
>
> Yes, we input specific $(x, y)$ pairs targeted for unlearning into the NPO objective. Concretely, while standard SFT uses gradient descent to minimize the loss and learn the $x$-to-$y$ mapping (where $y$ is the direct answer without reasoning in our experiments), NPO operates in reverse by introducing a negative loss function for these specific pairs, effectively applying gradient ascent. This penalizes the generation of the memorized answer $y$ given $x$, forcing the model to reduce its reliance on retrieval shortcuts.
>
> ---
>
> > Unify b_x (line 4) and x (line 9) in Algorithm 1 for easier to understand.
>
>
> We have revised line 4 in Algorithm 1 as “x and y”.
>
> ---
>
> Again, we thank the reviewer for the valuable feedback. Please let us know if there are any other questions or suggestions.
>
> Best,
>
> Authors

---

### Official Review · Reviewer_4tDC · 2025-11-04

**Soundness:** 3
**Presentation:** 3
**Contribution:** 3
**Rating:** 6
**Confidence:** 4

**Summary:**

This paper focuses on the two primary competing capabilities that influence the final answers of LRMs: deliberate reasoning through CoT and direct retrieval from internal memory. And the paper identifies the key factors that determine the dominance of reasoning versus retrieval through controlled intervention experiments, and ultimately proposes a post-training method, FARL, to regulate the relative strength of these two capabilities.

**Strengths:**

1. The competitive interplay between reasoning mechanisms and memory retrieval in LRMs is an important and timely research topic.

2. The paper is well-structured and clearly organized.

3. I like the idea and design of FARL — it effectively validates the paper’s conclusions and serves as a good contribution to the study.

**Weaknesses:**

1. In Section 3.1, I am considering whether SFT can serve as a reasonable method for directly modifying a model’s memory. While SFT does increase the probability of target tokens during deep-layer processing, evidence from mechanistic interpretability research [1, 2] suggests that it does not directly alter the MLP-stored factual knowledge or modify the model’s internal retrieval mechanisms. Therefore, the reliability of SFT as a means of memory intervention has a direct impact on the validity of the experimental conclusions in this work.

2. The investigation of the reasoning mechanism in LRMs is limited to injecting misleading cues into the CoT and observing the model’s response across domains. However, the paper does not examine the **intrinsic** reasoning behavior of LRMs within those domains. For instance, in mathematical tasks, the interplay between the model’s **inherent** mathematical reasoning and its memory recall mechanism is not explored.

3. Similarly, the study of retrieval mechanisms introduces new “memory” through SFT, rather than examining the model’s inherent knowledge retrieval capabilities.

4. The experiments are restricted to multiple-choice tasks. I would like to see results on open-ended generation settings as well, as such tasks would more naturally reflect the model’s reasoning and retrieval interplay in real-world scenarios.

5. I appreciate the problem studied in this paper. Although I guess some inspiration may come from *Competition of mechanisms: Tracing how language models handle facts and counterfactuals*, I see this as acceptable and good for me. Still, consistent with Point 4, I would like to see results across more diverse task formats to strengthen the conclusions.

Overall, I appreciate the idea presented in this work and would be happy to reconsider my rating if the above concerns are addressed sufficiently.

**Questions:**

Please see the weaknesses above.

---

> ### Author Response · Authors · 2025-11-20
>
> We thank the reviewer for the valuable feedback on improving this paper! Please find below our response to the reviewer’s questions.
>
> ---
>
> > In Section 3.1, I am considering whether SFT can serve as a reasonable method for directly modifying a model’s memory. While SFT does increase the probability of target tokens during deep-layer processing, evidence from mechanistic interpretability research [1, 2] suggests that it does not directly alter the MLP-stored factual knowledge or modify the model’s internal retrieval mechanisms. Therefore, the reliability of SFT as a means of memory intervention has a direct impact on the validity of the experimental conclusions in this work.
>
> While most existing mechanistic editing methods [1,3] are designed to modify facts (i.e., subject-relation-object triples) stored in LLMs, in this work, we need to edit general question-answer mappings. This distinction renders mechanistic editing methods less applicable. We therefore adopt supervised fine-tuning (SFT), a general yet effective method for modifying internal knowledge that enables us to test whether edited knowledge can be successfully retrieved at inference time. This choice is well-justified: recent work [1,4] shows that SFT achieves competitive performance with specialized mechanistic editing methods such as ROME [1] and MEMIT [3] on standard knowledge editing benchmarks, establishing it as a rigorous baseline for our investigation.
> To empirically validate the reliability of SFT-based editing, following prior work, we measure its performance using three key metrics [4]:
>
> **Efficacy.**  Let $x$ denote a question and $y^\ast$ be its edited answer. Efficacy measures the probability that model $\mathcal{M}$ produces $y^\ast$ as the answer to $x$, i.e., $y^\ast = \arg\max_y \mathcal{M}_\theta(y \mid x)$.  To avoid the influence of the model’s reasoning capability, we extract the answer directly using the prompt *"The correct answer is ("* without any chain-of-thought.  According to the table below, the majority of samples across all domains can be successfully perturbed, indicating that SFT effectively alters the LRM’s memory.
>
> |                        | MathLogic | SciEng | Computing | LifeSci | Health | BusinessEcon | Society | Humanities | Arc_Easy | Arc_Challenge |  GPQA |
> |------------------------|:---------:|:------:|:---------:|:-------:|:------:|:------------:|:-------:|:----------:|:--------:|:-------------:|:-----:|
> | Efficacy |   0.873   |  0.891 |    0.901    |  0.923  |  0.913  |     0.897    |   0.892   |    0.934   |   0.914  |      0.934      | 0.928 |
>
> **Generalization.** We further examine the robustness of the editing to question variation by measuring generalization: the probability that model $\mathcal{M}$  produces $y^\ast$ as the answer to a paraphrased question $\tilde{x}$: $y^\ast = \arg\max_y\mathcal{M}_\theta( y|\tilde{x})$. To this end, we randomly sample 50 successfully perturbed examples from each domain in the perturbation-to-retrieval experiment and generate paraphrased questions using GPT-5-mini. As shown in the Table below, the generalization measures remain high across domains, indicating that the editing exhibits strong robustness to question paraphrasing.
>
> |                        | MathLogic | SciEng | Computing | LifeSci | Health | BusinessEcon | Society | Humanities | Arc_Easy | Arc_Challenge |  GPQA |
> |------------------------|:---------:|:------:|:---------:|:-------:|:------:|:------------:|:-------:|:----------:|:--------:|:-------------:|:-----:|
> | T-PSR (Rewrite Prompt) |   0.975   |  0.925 |    0.9    |  0.925  |  0.95  |     0.875    |   0.9   |    0.925   |   0.950  |      0.9      | 0.925 |
>
> **Locality.** We also assess whether SFT-based editing inadvertently affects unrelated knowledge by measuring locality. Formally, locality quantifies the probability that $\mathcal{M}$ generates the ground-truth answer $y'$ for an unrelated question $x'$: $y' = \arg\max_y\mathcal{M}_\theta( y|x')$. To evaluate this, we fine-tune the model exclusively on the MathLogic domain using incorrect answers and measure performance on all samples in other domains both before and after SFT. The table below compares the locality before and after SFT under two settings: CoT enabled and CoT disabled. In both cases, the locality after SFT is almost identical to that before SFT, suggesting that the intervention produces targeted, localized changes rather than inducing global knowledge degradation.
>
> |  | Before SFT | After SFT |
> |:-----------:|:-----------:|:-----------:|
> | w/o CoT | 0.261 | 0.263 |
> | w/ CoT | 0.713 | 0.716 |
>
> ---
> [1] Meng et al., Locating and Editing Factual Associations in GPT, NeurIPS’22. \
> [2] Zhang et al., A Comprehensive Study of Knowledge Editing for Large Language Models, Arxiv.\
> [3] Meng et al., Mass-editing memory in a transformer, ICLR’23.\
> [4] Gangadhar et al., Model Editing by Standard Fine-Tuning, ACL’24.

---

> ### Author Response · Authors · 2025-11-20
>
> > The investigation of the reasoning mechanism in LRMs is limited to injecting misleading cues into the CoT and observing the model’s response across domains. However, the paper does not examine the intrinsic reasoning behavior of LRMs within those domains. For instance, in mathematical tasks, the interplay between the model’s inherent mathematical reasoning and its memory recall mechanism is not explored.
>
> To answer the reviewer’s question, we study the dynamics of retrieval-reasoning interaction at each reasoning step through the logit lens (details in Appendix D.2 of the revised paper).
>
> We track the logits of the retrieval-led answer $y_t$ and the reasoning-led answer $y$ throughout the model's reasoning process. We employ the perturbation-to-retrieval setting (Section 3.1), where the model has been fine-tuned via SFT to associate the question with an incorrect answer $y_t$, while the reasoning leads to the correct answer $y$. We split the model's generated reasoning into discrete steps (delimited by *"\n\n"*). We then progressively prefill the context with each reasoning step and probe the model's answer distribution at each step using the prompt *``The correct answer is (''*. Figure 11 illustrates a dynamic competition between retrieval and reasoning:
>
> - Reasoning-Dominant Cases: Figures 11(a) and 11(b) present two representative examples where the model converges on the reasoning-led answer $y$. In both cases, the logit for the perturbed answer $y_t$ begins high but steadily declines as reasoning progresses. Conversely, the logit for $y$ starts low but increases monotonically throughout the reasoning chain, eventually suppressing $y_t$.
>
> - Retrieval-Dominant Cases: In contrast, Figures 11(c) and 11(d) show two representative examples where the model produces the retrieval-led answer $y_t$. Here, the logit of $y_t$ remains higher than $y$ throughout the entire reasoning process, indicating that retrieval consistently overrides the reasoning pathway.
>
> These fine-grained logit trajectories provide direct evidence of the dynamic interaction between retrieval and reasoning at each step of the reasoning process. The divergent patterns across examples support our claim that these two pathways tend to compete with each other in an intricate manner. This analysis suggests that our study reflects genuine retrieval-reasoning interaction.
>
> ---
>
> > Similarly, the study of retrieval mechanisms introduces new “memory” through SFT, rather than examining the model’s inherent knowledge retrieval capabilities.
>
> We perform a causal intervention experiment based on activation patching [2] to verify that the attention heads identified in our attention pattern analysis (Section 3.4) causally control the selection between the two pathways (more details in Appendix D.3). The experiment is conducted on R1-Llama-8B using the Math&Logic domain of the MMLU dataset.
>
> Under the perturbation-to-retrieval setting, we collect attention activations from (i) successfully perturbed samples in the SFT-tuned model (perturbation run) and (ii) their corresponding activations in the base model (clean run). We then conduct two complementary interventions:
>
> - Replacing perturbation-run activations with clean-run activations for the attention heads ranked in the top 5% by AUC score (Figure 5) restores the original answers with 87.2% success rate. The same intervention on an equal number of randomly selected attention heads yields only 5.3% restoration.
>
> - The reverse intervention, patching clean-run activations with perturbation-run activations, produces perturbed answers with 89.5% success rate for the top 5% AUC heads, but only 2.1% for randomly selected heads.
>
> These results indicate that high-AUC attention heads exert causal control over the competition between reasoning and retrieval pathways, with the final answer exhibiting clear counterfactual dependence on their activation patterns.

---

> ### Author Response · Authors · 2025-11-20
>
> > The experiments are restricted to multiple-choice tasks. I would like to see results on open-ended generation settings as well, as such tasks would more naturally reflect the model’s reasoning and retrieval interplay in real-world scenarios.
>
> To validate the generalizability of our conclusion to non-multiple-choice QA, we use GeneralThought [5], a free-form QA dataset that spans diverse question types and domains (more details in Appendix D.4 of the revised paper). We randomly sample 400 questions with answers that are relatively easy to verify (e.g., numbers, nouns, short phrases). We then use GPT-5-mini to produce the misleading answers $y_r$ and $y_t$ and run the perturbation pipeline introduced in Section 3.1. The table below reports the joint influence of reasoning and retrieval perturbations, showing that their effects closely match those observed on the multiple-choice QA dataset (e.g., Figure 2).
>
> | Model \ Metric | T-PSR | R-PSR | T-PSR + R-PSR (Aligned Answers) | R-PSR (Disparate Answers) | T-PSR (Disparate Answers) |
> |----------------|:-----:|:-----:|:-------------------------------:|:-------------------------:|:-------------------------:|
> | R1-Llama-8B    | 0.320 | 0.563 |              0.652              |           0.458           |           0.181           |
> | R1-Qwen-7B     | 0.292 | 0.642 |              0.772              |           0.121           |           0.552           |
> | Phi4-mini      | 0.095 | 0.352 |              0.431              |           0.345           |           0.185           |
> | Qwen3-8B       | 0.145 | 0.179 |              0.340              |           0.277           |           0.093           |
>
> ---
> > I appreciate the problem studied in this paper. Although I guess some inspiration may come from Competition of mechanisms: Tracing how language models handle facts and counterfactuals, I see this as acceptable and good for me. Still, consistent with Point 4, I would like to see results across more diverse task formats to strengthen the conclusions.
>
> We appreciate the reviewer's insightful comments. While both studies examine how different mechanisms shape final answers, our work differs in several key aspects. First, [6] studies how LLMs process conflicting information between factual knowledge and counterfactual context, whereas we study whether LRMs rely on CoT reasoning or retrieve memorized answers. Second, [6] constructs counterfactuals within the prompt, while we perturb both reasoning chains (by injecting misleading cues) and memory (through SFT). Finally, we translate our attribution findings into FARL, a method that improves CoT robustness by suppressing retrieval shortcuts.
>
> ---
>
> [5] GeneralThought dataset: huggingface.co/datasets/RJT1990/GeneralThoughtArchive \
> [6] Ortu et al., Competition of mechanisms: Tracing how language models handle facts and counterfactuals, ACL’24.
>
> ---
>
> Again, we thank the reviewer for the valuable feedback. Please let us know if there are any other questions or suggestions.
>
> Best,
>
> Authors

---

### Author Response · Authors · 2025-12-02
**Rebuttal Summary**

Dear AC, SAC, and PCs,

Thank you for your time and effort in reviewing our paper. To provide a comprehensive view of our work and responses, we include below (1) an overview of our paper, (2) a summary of reviewers’ comments and updates, and (3) a brief walkthrough of the rebuttal process.

---

**[Overview]**

This work introduces a perturbation-based evaluation framework showing that large reasoning models  (LRMs) derive final answers through two competing mechanisms, direct memory retrieval and Chain-of-Thought (CoT) reasoning, and their relative dominance varies across domains, model scales, and causally controllable attention heads. Building on these insights, we propose FARL (forgetting-augmented reinforcement learning), a novel framework that suppresses retrieval shortcuts and strengthens robust, generalizable learning of reasoning capabilities.

---

**[Reviewers’ Comments and Updates]**

- Reviewer 4tDC is **positive with high confidence**. Their major questions concern the effectiveness of memory editing by SFT and competing dynamic analysis. **They noted willingness to raise their score if concerns are addressed.**

- Reviewer fSyB finds our work methodologically rigorous and impactful. They requested evidence on open-ended questions and causal analysis, to which we provide additional experiments to demonstrate the robustness of our findings across diverse question types and establish causal relationships through controlled ablation studies.

- Reviewer kodm is **positive with high confidence**. Their major questions concern whether SFT harms reasoning ability and requested evidence on open-ended questions. **They were satisfied with our rebuttal and maintained their positive score.**

- Reviewer 3Cxz appreciates our broad and well-designed experiments. Their main question is whether SFT harms reasoning ability. **They noted willingness to raise their score if concerns are addressed.**

---

**[Rebuttal Walkthrough]**

We have provided point-by-point responses to all reviewers' questions, with all revisions highlighted in the revised manuscript. Below is a summary of the major questions we have addressed.

**1. Validation of SFT-based memory editing (Reviewer 4tDC, kodm, 3Cxz)**

Reviewers requested stronger evidence that SFT performs localized memory editing without degrading reasoning. We validated SFT-based editing using three standard knowledge editing metrics—efficacy (87–93% perturbation success rate on retrieval), generalization (87–97% perturbation success rate when questions are rewritten), and locality (near-identical accuracy before 0.713 and after 0.716 on invariant domains), confirming targeted edits with preserved global capability (Appendix D.1).

**2. Intrinsic reasoning vs. retrieval behavior analysis (Reviewer 4tDC, fSyB, kodm)**

Reviewers encouraged deeper introspective analysis to differentiate retrieval-dominant and reasoning-dominant pathways. We expanded introspective analysis by adding a token-level logit lens study (Appendix D.2, Fig. 11), revealing dynamic transitions between retrieval-dominant and reasoning-dominant pathways across reasoning steps.

**3. Causal verification of competing mechanisms (Reviewer 4tDC, fSyB, kodm)**

To prove causal effects of the attention heads identified in RQ2, we conducted activation patching experiments (Appendix D.3). Results show that the top 5% AUC-ranked heads show 87.2% / 89.5% causal intervention success rate in switching reasoning or retrieval-dominant behavior, far higher than random heads (~2–5%).

**4. Open-ended question validation (Reviewer 4tDC, fSyB, kodm)**

Reviewers asked whether our conclusions generalize beyond multiple-choice settings. We extended evaluation to free-form QA on the 400-sample GeneralThought dataset (Appendix D.4), finding that retrieval–reasoning dominance patterns remain consistent with the MCQ setting.

---

**[Final Remark]**

We believe our rebuttal and additional experiments have thoroughly addressed the reviewers' questions. Please let us know if you have any further questions or suggestions.

Best,

Authors

---

### Meta-Review · Area_Chair_fQ28 · 2026-01-24

**Summary:**

Reviewers found the problem timely and the paper generally clear. Key concerns were whether SFT is a valid way to “edit memory,” whether R-PSR/T-PSR support causal claims (vs. correlations), limited evidence beyond multiple-choice QA, and how well the attention-head analysis and FARL are motivated and connected to the main claims.

**Reviewer Concerns:**

Addressed:

* Added free-form QA evidence (GeneralThought), reducing the “MCQ-only” concern.

* Added causal intervention via activation patching for high-AUC heads.

* Added token/logit-level dynamics (logit lens over steps), beyond binary R-PSR/T-PSR.

* Added SFT-edit validation (efficacy/generalization/locality) and a cue-position ablation to argue effects are not just artifacts.

Still outstanding / partial:

* “Retrieval” is still operationalized mainly as question→answer shortcut editing via SFT; the link to intrinsic factual-retrieval circuitry is plausible but not fully nailed down.

* Mechanistic language remains somewhat strong given the limited scale/architectures tested.

* FARL’s gains come with longer CoTs and limited training scope; broader generality is still uncertain.

**Reviewer Scores:**

* 4tDC (6 → ~7): likely up with open-ended + causal + SFT validation.

* fSyB (2 → ~5/6): major asks were met; could still be cautious on framing/claims.

* kodm (6 → 6): indicated they would keep the positive rating.

* 3Cxz (4 → ~5/6): key plausibility concerns were largely addressed.

---

### Decision · Program_Chairs · 2026-01-26

Accept (Poster)